# An iron-dependent metabolic vulnerability underlies VPS34-dependence in RKO cancer cells

**Marek J. Kobylarz**[1], **Jonathan M. Goodwin**[1¤a], **Zhao B. Kang**[1], **John W. Annand**[1¤a], **Sarah Hevi**[1], **Ellen O'Mahony**[1], **Gregory McAllister**[1¤b], **John Reece-Hoyes**[1], **Qiong Wang**[1], **John Alford**[1], **Carsten Russ**[1], **Alicia Lindeman**[1], **Martin Beibel**[2], **Guglielmo Roma**[2], **Walter Carbone**[2], **Judith Knehr**[2], **Joseph Loureiro**[1], **Christophe Antczak**[1], **Dmitri Wiederschain**[1¤c], **Leon O. Murphy**[1¤a], **Suchithra Menon**[1]*, **Beat Nyfeler**[2]*

**1** Novartis Institutes for Biomedical Research, Cambridge, MA, United States of America, **2** Novartis Institutes for Biomedical Research, Basel, Switzerland

¤a  Current address: Casma Therapeutics, Cambridge, MA, United States of America
¤b  Current address: Sana Biotechnology, Cambridge, MA, United States of America
¤c  Current address: Sanofi, Cambridge, MA, United States of America
*  sue.menon@novartis.com (SM); beat.nyfeler@novartis.com (BN)

**Data Availability Statement:** RNA-seq data are available from the NCBI Sequence Read Archive (accession number PRJNA633293).

**Funding:** The funder, Novartis Pharma AG, provided support in the form of salaries for authors

## Abstract

VPS34 is a key regulator of endomembrane dynamics and cargo trafficking, and is essential in cultured cell lines and in mice. To better characterize the role of VPS34 in cell growth, we performed unbiased cell line profiling studies with the selective VPS34 inhibitor PIK-III and identified RKO as a VPS34-dependent cellular model. Pooled CRISPR screen in the presence of PIK-III revealed endolysosomal genes as genetic suppressors. Dissecting VPS34-dependent alterations with transcriptional profiling, we found the induction of hypoxia response and cholesterol biosynthesis as key signatures. Mechanistically, acute VPS34 inhibition enhanced lysosomal degradation of transferrin and low-density lipoprotein receptors leading to impaired iron and cholesterol uptake. Excess soluble iron, but not cholesterol, was sufficient to partially rescue the effects of VPS34 inhibition on mitochondrial respiration and cell growth, indicating that iron limitation is the primary driver of VPS34-dependency in RKO cells. Loss of RAB7A, an endolysosomal marker and top suppressor in our genetic screen, blocked transferrin receptor degradation, restored iron homeostasis and reversed the growth defect as well as metabolic alterations due to VPS34 inhibition. Altogether, our findings suggest that impaired iron mobilization via the VPS34-RAB7A axis drive VPS34-dependence in certain cancer cells.

## Introduction

VPS34 is a class III PI3-kinase that is ubiquitously expressed across eukaryotes and essential for physiology and organismal development. Perturbations of VPS34 result in pleiotropic defects by altering vesicular trafficking [1–4], intracellular signaling [5–7], metabolism [8, 9] and cellular growth [10, 11]. Mice with knockout of *Vps34* are embryonic lethal arresting at

MJK, JMG, ZBK, JWA, SH, EO, GM, JR, QW, JA, CR, AL, MB, GR, WC, JK, JL, CA, DW, LOM, SM, and BN, but did not have any additional role in the study design, data collection and analysis, or preparation of the manuscript. The manuscript was assessed by the Novartis legal department to ascertain that release of this manuscript did not contravene Novartis policy regarding release of proprietary information, but this was limited only to approval for release of the material and was not related to the results or interpretations/conclusions contained in the manuscript.

**Competing interests:** The authors are, at present, or were during the time of their contribution to this manuscript, employed by Novartis Pharma AG. As such, the authors received salaries and own stock in Novartis as part of their remuneration for employment. There are no competing interests as regards consultancies, patents or products in development or currently marketed. This does not alter the authors' adherence to all the PLoS ONE policies on sharing data and materials.

gastrulation stage [10]. Partial inactivation of VPS34, using a heterozygous kinase-dead mouse model, was found to trigger a metabolic switch from oxidative phosphorylation to glycolysis and enhance insulin sensitivity as well as glucose tolerance *in vivo* [8].

VPS34 catalyzes the conversion of phosphatidylinositol to phosphatidylinositol-(3)-phosphate (PI(3)P), which serves as a membrane marker to recruit effector proteins with PI(3)P binding domains, such as FYVE, or PX domains [12–16]. Through modulation of the subcellular localization and/or function of these effector proteins, VPS34 regulates autophagy and endocytic trafficking [17–20]. VPS34 functions in at least two protein complexes, termed complex 1 (C1) and complex 2 (C2). C1 contains VPS34-VPS15-BECN1-ATG14 and regulates the initiation of autophagy while C2 consists of VPS34-VPS15-BECN1-UVRAG and controls endosome and phagosome maturation [21, 22].

The Cancer Dependency Map (www.depmap.org, [23]) classifies VPS34 as common essential gene based on CRISPR-mediated depletion data across several hundred cancer cell lines. To determine which function of VPS34 drives dependency in cancer cells, we carried out a series of studies using PIK-III, a potent and selective inhibitor of VPS34 [1] to control the degree and timing of VPS34 inhibition. We discovered a strong VPS34-dependence in RKO cells, which was further characterized by a genetic suppressor screen and transcriptional profiling. Our data shows that acute inhibition of VPS34 enhanced the lysosomal degradation of transferrin receptor in a RAB7A-dependent manner, thereby starving cells of iron, an essential nutrient. We found that excess iron was sufficient to partially rescue impaired mitochondrial respiration and cell proliferation upon VPS34 inhibition in RKO cells, highlighting a key role for VPS34 in regulating iron availability for metabolism and cell growth.

## Results

### A genome-wide CRISPR screen identifies endolysosomal pathway components as genetic suppressors of VPS34 vulnerability

To identify potential VPS34 vulnerabilities in cancer cells we assessed cell growth after a 3 day treatment with the VPS34 inhibitor PIK-III across 306 cell lines from the Cancer Cell Line Encyclopedia (CCLE) [24]. Among the cell lines that demonstrated reduced growth in the presence of PIK-III, RKO colorectal cancer cells were particularly sensitive to VPS34 inhibition (Fig 1A). Furthermore, a large-scale pooled shRNA screen across the CCLE (project DRIVE, [25]) also identified RKO as strongly sensitive to VPS34, VPS15 and BECN1 knockdown (Fig 1B). RKO was also one of the most sensitive cell lines to the knockdown of VPS34 complex C2 component UVRAG while shRNAs against VPS34 complex C1 component ATG14 did not robustly affect RKO cell growth (Fig 1B). In addition, project DRIVE data (S1A Fig) and individual dox-inducible shRNAs (S1B and S1C Fig) showed that knockdown of canonical autophagy genes ATG5, ATG7 and ATG16L1 do not significantly impact growth of RKO cells. To confirm VPS34-dependency in RKO cells we expressed a doxycycline-inducible VPS34 shRNA to deplete VPS34 (Fig 1C). Knockdown of VPS34 decreased RKO cell growth in both colony formation and cell proliferation assays and this sensitivity was further enhanced under low serum condition (Fig 1D and 1E). Overexpression of shRNA-resistant wild-type VPS34 (WT), but not a kinase dead (KD) variant, was able to rescue cell growth (Fig 1C–1E). Together, these data define RKO as a cell line depending on the catalytic activity of VPS34, specifically in the C2 complex.

We next took an unbiased approach to identify genetic modifiers of VPS34 vulnerability by performing a proliferation-based CRISPR screen. For this purpose, Cas9-expressing RKO cells were infected with a whole-genome sgRNA library and treated for 28 days in the presence or absence of 0.5 μM PIK-III, a low dose slightly above the cellular IC50 (Fig 2A). This screening

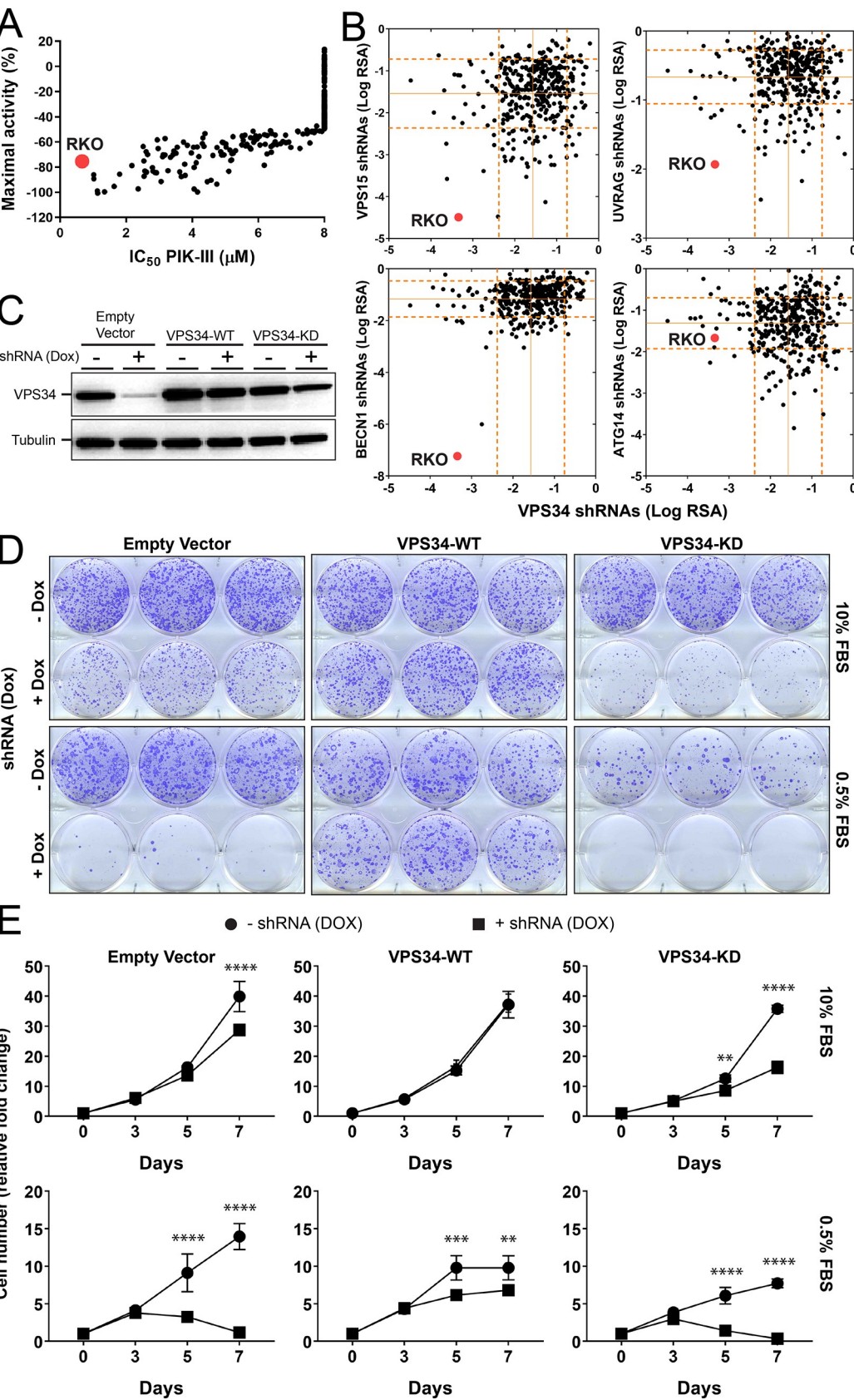

**Fig 1. VPS34-dependency in the colorectal cancer cell line RKO.** (A) Cell line profiling. 306 cancer cell lines were treated with PIK-III and cell proliferation was measured after 72 hours using the CellTiter-Glo assay. PIK-III IC50 and maximal activity values were determined relative to vehicle control (0%) and MG132 treatment (100%). Each dot represents a cell line and RKO is highlighted in red. The entire dataset is reported in S1 Table. (B) Pooled shRNA screening data for VPS34, VPS15, BECN1, ATG14 and UVRAG was extracted from project DRIVE [25] and visualized as RSA significance scores. Each dot represents a cell line and RKO is highlighted in red. The straight and dotted lines indicate average and 2x standard deviation of the RSA values across all cell lines, respectively. (C-E) Empty vector, shRNA resistant wild type (VPS34-WT) or kinase dead variant VPS34 constructs (VPS34-KD) were introduced into RKO cells expressing doxycycline (Dox) inducible *VPS34* shRNA. Cells were grown in medium with 0.5% or 10% fetal bovine serum (FBS). After Dox-induced VPS34 knockdown, cell lysates were probed by immunoblotting (C), cell colony formation by crystal violet staining (D), and cell proliferation by CellTiter-Glo assay (E). For CellTiter-Glo, the average of three independent experiments is shown and error bars represent the standard deviation (SD). **, $p < 0.01$; ***, $p < 0.001$; ****, $p < 0.0001$ (two-way ANOVA).

paradigm identified many suppressors of PIK-III, which are CRISPR gene knockouts (KO) rescuing RKO cell proliferation in the presence of the VPS34 inhibitor (Fig 2B). Gene ontology enrichment analysis (Molecular Signatures Database [26, 27]) of PIK-III suppressors with a

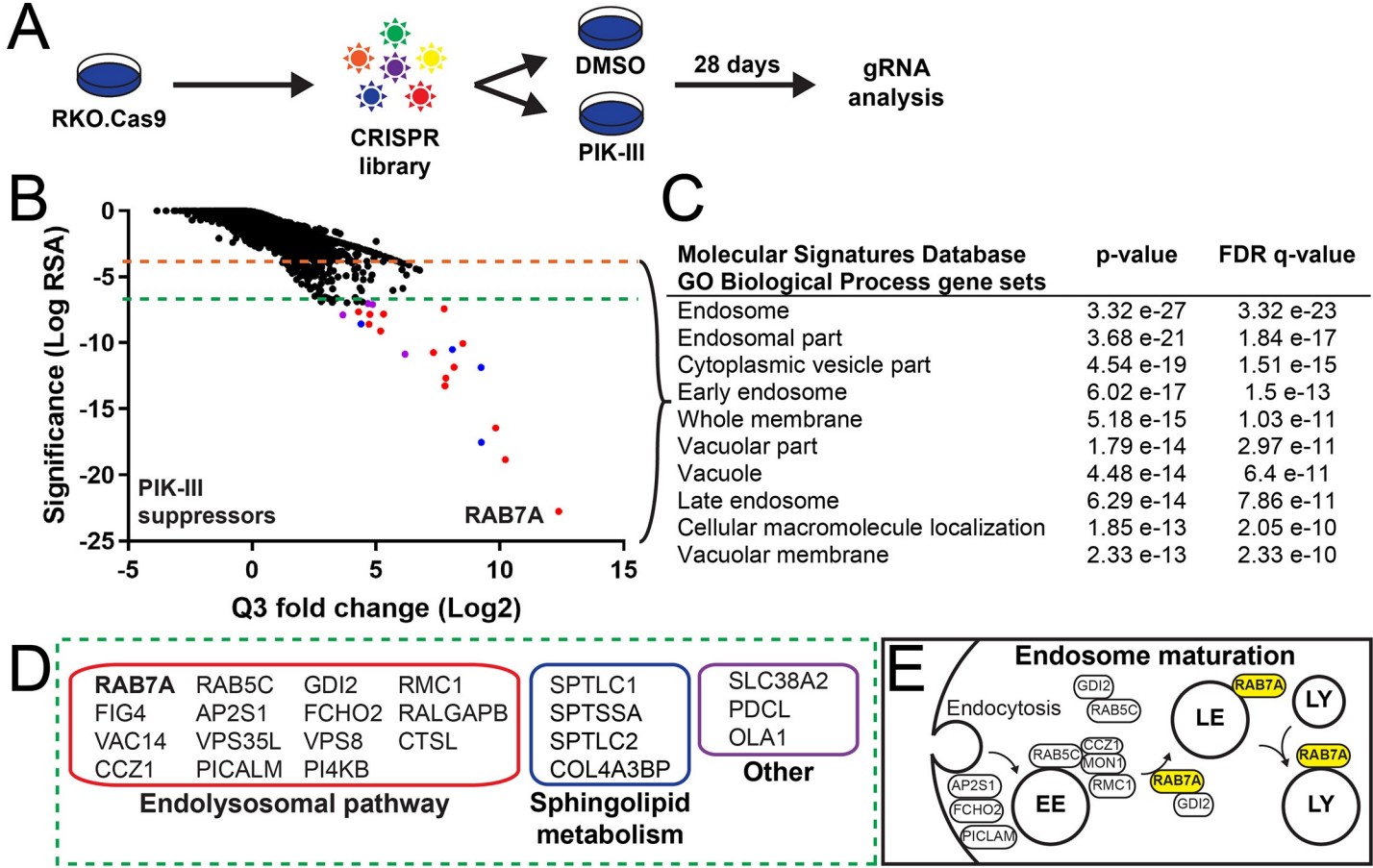

**Fig 2. A genome-wide CRISPR screen identifies endosomal maturation pathway genes as suppressors of PIK-III.** (A) Schematic of the genome-wide pooled CRISPR screening workflow for genetic modulators of VPS34 vulnerability. (B) Gene-centric visualization of the significance and Q3 fold change of sgRNAs detected in cells treated with 0.5 μM PIK-III versus vehicle control. The entire dataset is reported in S2 Table. Significance cutoff values -4 and -7 are colored in orange and green dotted lines, respectively. (C) Gene Ontology (GO) biological process enrichment analysis of PIK-III suppressor hits below the significance cutoff value of -4. The 10 most significant GO terms are shown. (D) The strongest PIK-III suppressor hits below the significance cutoff value of -7 were categorized into groups labeled the Endolysosomal pathway (red box), Sphingolipid metabolism (blue box) and Other (purple box). (E) Schematic of the endosome to lysosome transport pathway (endosome maturation) reconstituted from hits derived from the CRISPR screen. RAB7A, the top PIK-III suppressor hit, is highlighted in yellow. EE, LE and LY stand for early endosome, late endosome and lysosome, respectively.

significance RSA cutoff value of ≤ -4.0 highlighted endosomal and vacuolar biological processes (Fig 2C). Looking more closely at some of the strongest PIK-III suppressor hits with a significance RSA cutoff value of ≤ -7.0, we identified sphingolipid metabolism genes (*SPTLC1*, *SPTLC2* and *SPTSSA*) as well as genes involved in endosomal-lysosomal maturation *(RAB5C, RAB7A, CCZ1* and *RMC1)*, phosphatidylinositol modification (*VAC14* and *FIG4*), clathrin-dependent endocytosis (*AP2S1*), early endosome tethering (*VPS8*), lysosomal degradation *(CTSL)*, and a retriever complex component (*VPS35L*) (Fig 2D). Many of the top hits reconstitute the endosome to lysosome transport pathway (endosome maturation), which is necessary for cargo delivery to lysosomes for degradation (Fig 2E). Based on this data, we hypothesized that PIK-III impairs RKO cell growth by deregulating trafficking within the endosome to lysosome transport pathway.

## VPS34 inhibition alters cholesterol homeostasis and hypoxia response

In order to characterize the molecular changes following VPS34 inhibition we carried out a transcriptional profiling study upon PIK-III treatment in RKO cells. After 24 hours treatment, 189 genes were induced and 7 genes repressed more than 2-fold by PIK-III (Fig 3A). An enrichment analysis of the upregulated genes using the Hallmark gene sets from the Molecular Signatures Database [26–28] revealed cholesterol homeostasis and hypoxia as the top two differentially regulated gene signatures (Fig 3B and 3C). Consistent with the enrichment analysis, PIK-III induced Sterol Regulatory Element-Binding Protein 2 (SREBP2) cleavage and Hypoxia Inducible Factor 1α (HIF1α) levels, two transcriptional master regulators of cholesterol metabolism and hypoxia response, respectively (Fig 3D and 3E). Deprivation of intracellular cholesterol triggers SREBP2 cleavage and translocation of processed SREBP2 to the nucleus to turn on cholesterol biosynthetic genes [29]. HIF1α is regulated by oxygen via iron-containing prolyl hydroxylases, which promote the proteasomal degradation of HIF1α. Deprivation of oxygen, or iron stabilizes HIF1α, which in turn induces hypoxia response genes [30, 31]. The induction of SREBP2 and HIF1α suggests that PIK-III treatment may deplete intracellular cholesterol and iron, as oxygen is not limiting in our system.

## VPS34 regulates cholesterol uptake through lysosomal clearance of LDL receptor

To address if PIK-III depletes intracellular cholesterol, we examined cholesterol levels by lipid profiling. The lipidomic analysis confirmed severe reduction of sterol lipid species comprised of zymosteryl esters, cholesteryl esters and free cholesterol in PIK-III-treated RKO cells (Fig 4A and S4 Table). Cholesterol can be synthesized *de novo*, or taken up through endocytosis as part of low-density lipoproteins (LDL) by LDL receptor (LDLR) [29, 32]. Since many hits in our PIK-III suppressor screen regulate endosome to lysosome transport, we focused on the endocytosis step and monitor uptake of fluorescently labelled LDL. Endocytosis of the LDL probe was readily visible in vehicle-treated cells and a 24 hour treatment with PIK-III significantly impaired uptake (Fig 4B). Quantification revealed that PIK-III decreased the average LDL probe intensity by 35% (Fig 4C). We next asked whether LDL receptor level was impacted by VPS34 inhibition. Western blotting for LDLR showed that protein levels were strongly reduced by PIK-III in a dose and time-dependent manner (Fig 4D). The autophagy receptor p62 was used as a control to monitor VPS34-mediated autophagy inhibition, which also revealed a similar dose and time-dependent effect (Fig 4D). Since RNA sequencing data did not reveal any reduction in LDLR mRNA levels upon PIK-III treatment (S3 Table), we set out to determine if the decrease in LDLR level was due to enhanced lysosomal degradation using the v-ATPase inhibitor bafilomycin A1 to block lysosomal activity. Bafilomycin A1 fully

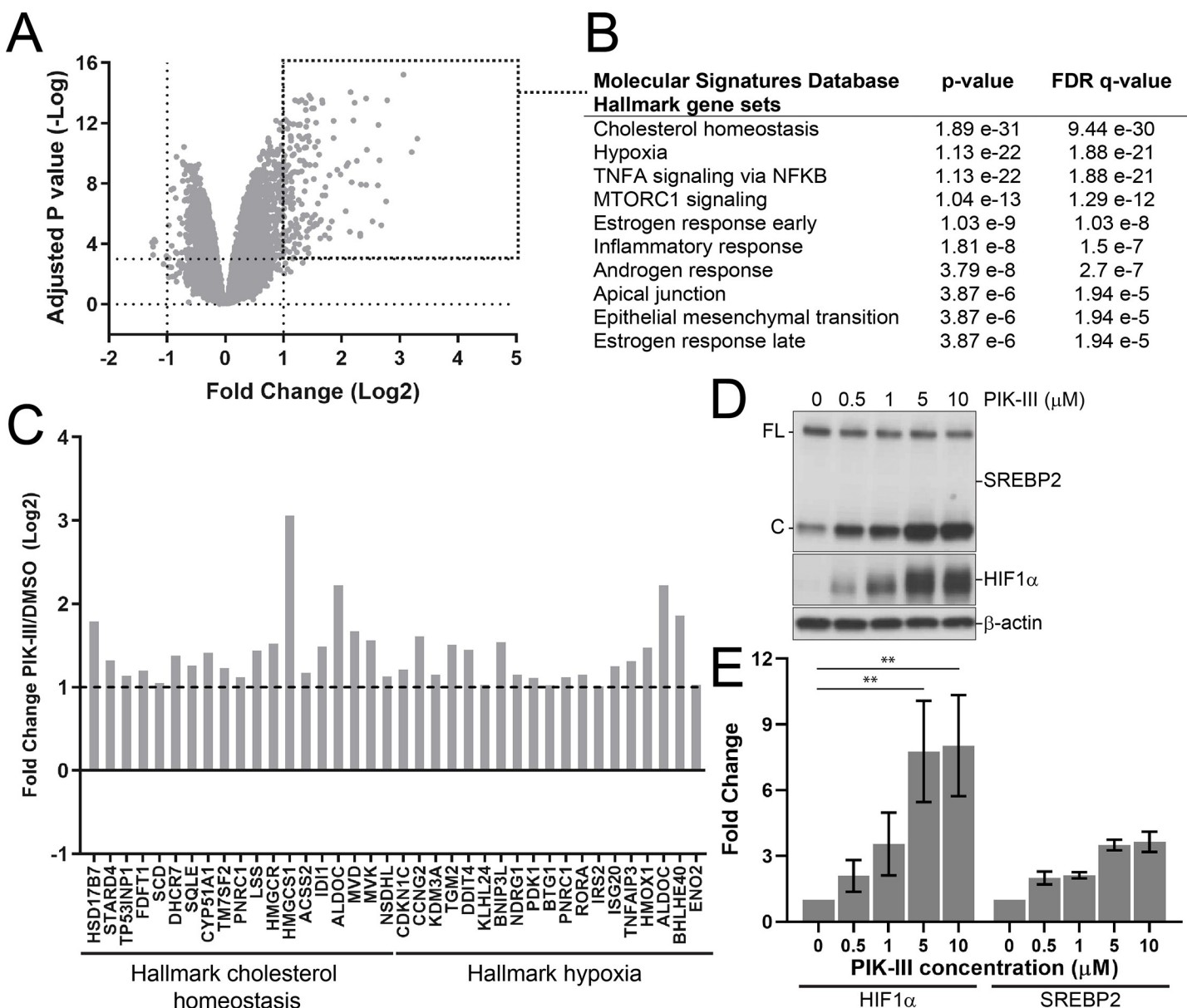

**Fig 3. Inhibition of VPS34 induces sterol regulatory and hypoxia target genes.** (A) Gene expression profile of RKO cells upon VPS34 inhibition. RKO CTRL cells were treated with 0.5 μM PIK-III or vehicle control for 24 hours and total RNA was extracted and subjected to sequencing. The volcano plot visualizes genes differentially expressed upon PIK-III treatment. Data represents the average of three replicates. The entire dataset is reported in S3 Table. (B) Gene set enrichment analysis was performed on genes induced by PIK-III in RKO cells with a fold change >2 and an adjusted p value < 0.001. The 10 most significant hallmark gene sets are shown. (C) Log2 expression fold changes for individual genes of the cholesterol homeostasis and hypoxia gene sets in RKO cells. (D) RKO CTRL cells were treated with the indicated concentrations of PIK-III for 24 hours and lysates were immunoblotted with the specified antibodies. Full-length *(FL)* and cleaved SREBP2 *(C)* are indicated. (E) Quantitation of immunoblot in (D) as fold change relative to vehicle treatment. All data are normalized to the β-actin loading control. Data is the mean of four independent experiments ± SD. **, p < 0.01 (two-way ANOVA).

prevented PIK-III-mediated reduction of LDLR (Fig 4E) and this was also observed upon lysosomal inhibition with chloroquine (Fig 4F). Finally, we asked if cholesterol deprivation contributed to the VPS34-dependent growth defect. Rescue experiments with exogenous cholesterol showed that addition of excess cholesterol was not sufficient to rescue the growth defect of RKO cells treated with PIK-III (Fig 4G). Therefore, while PIK-III mediated VPS34 inhibition leads to depletion of cellular cholesterol and impaired uptake of LDL via

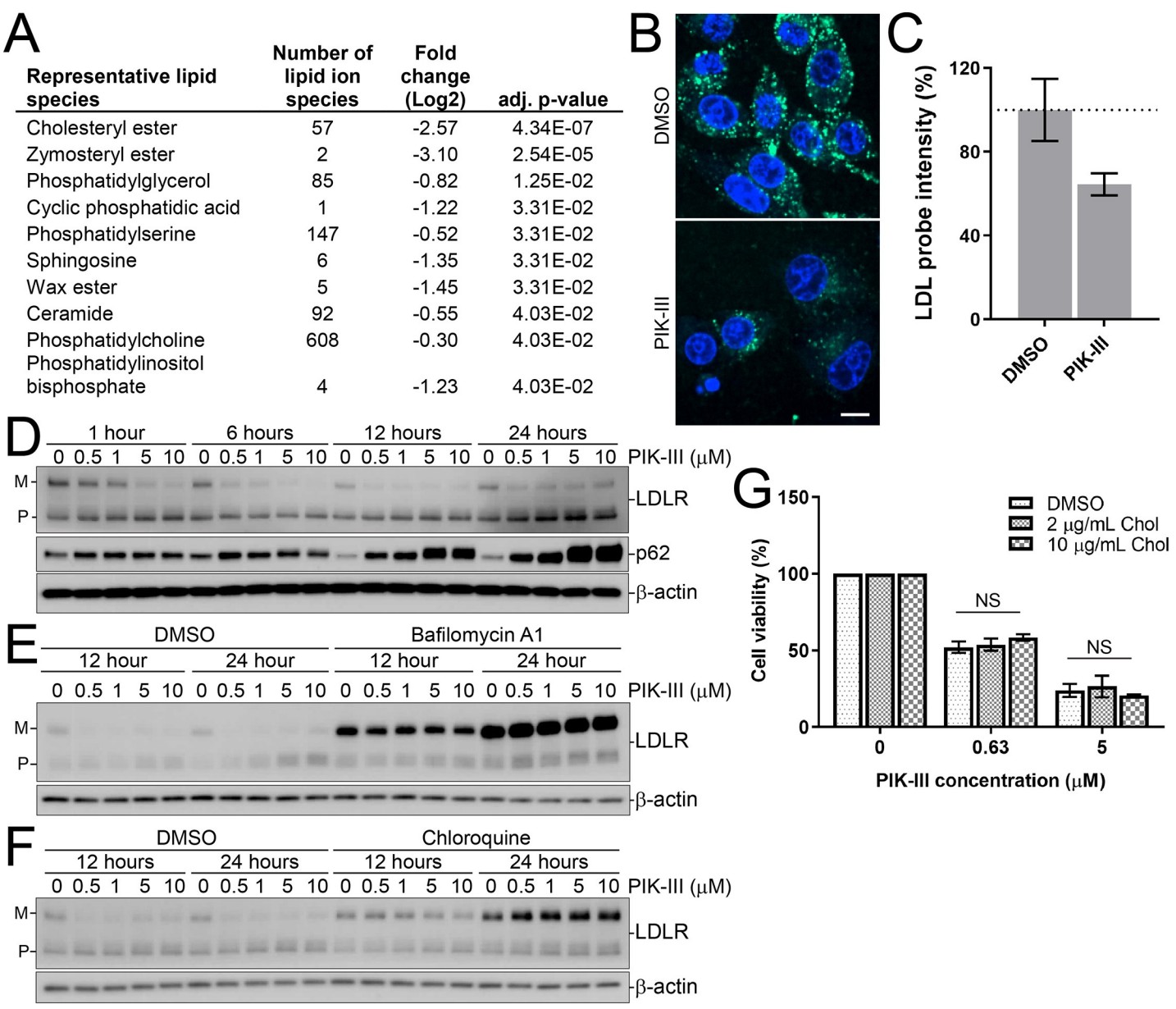

**Fig 4. VPS34 inhibition decreases intracellular levels of cholesteryl-esters, blocks cholesterol uptake and enhances lysosomal degradation of LDLR.** (A) Lipidomic analysis of RKO cells upon VPS34 inhibition. RKO CTRL cells were treated with 1 µM PIK-III for 24 hours and lipids were extracted and subjected to liquid chromatography mass spectrometry analysis. The table shows PIK-III-induced log2 fold change for aggregated and averaged lipid classes in RKO cells. The adjusted p-values were calculated between PIK-III and vehicle treatment group. Data represents the mean of three biological replicates and the entire dataset is reported in S4 Table. (B) RKO.Cas9 cells were treated with vehicle or 1 µM PIK-III for 24 hours followed by incubation with BODIPY FL LDL probe for 2 hours at 37˚C. Nuclei were labeled with Hoechst and live cell imaging was performed. The white bar represents 10 µm in length. (C) BODIPY FL LDL probe cellular uptake in (B) was quantified as percent average intensity of total cells. Data in the form of technical replicates were averaged and presented as the mean ± SD from 6 wells. (D) RKO CTRL cells were treated with the indicated concentrations of PIK-III for 1, 6, 12, or 24 hours. *(M)* stands for mature protein and *(P)* stands for precursor protein. (E, F) RKO CTRL cells were treated with the indicated concentrations of PIK-III with or without 50 nM bafilomycin A1 (E) or 30 µM chloroquine (F) for 12 and 24 hours. Lysates were immunoblotted with the specified antibodies. *(M)* and *(P)* indicate the mature and precursor protein, respectively. (G) RKO CTRL cells were treated with indicated concentrations of PIK-III along with vehicle, 2 µg/ml and 10 µg/ml soluble cholesterol (Chol) for 5 days and cell viability was assessed using the CellTiter-Glo assay. Data presented is the mean from three independent experiments ± SD. NS, not significant (two-way ANOVA).

enhanced degradation of LDLR, cholesterol deprivation is likely not the sole driver of PIK-III sensitivity.

## VPS34 regulates iron uptake through lysosomal clearance of transferrin receptor

The induction of HIF1α, which we observed upon PIK-III treatment (Fig 3D) may be caused by depletion of intracellular iron, as oxygen is not limiting in our system. Iron is an essential nutrient that plays a crucial role in many metabolic pathways such as hypoxia signaling, DNA synthesis, mitochondrial respiration and oxygen transport [33–36]. To confirm that VPS34 inhibition deprived RKO cells of iron, we ran an aconitase activity assay. Aconitase activity requires an iron-sulfur cluster as a cofactor and iron deprivation causes a defect in iron-sulfur cluster formation, leading to lower cofactor occupancies and a decrease in activity [37]. We observed robust inhibition of aconitase activity upon PIK-III treatment, similar to the iron chelator deferasirox (DFX) that was used as a control to set a baseline for severe iron deprivation (Fig 5A). Cellular deprivation of iron was further substantiated by assessing the levels of the Iron-Responsive element-binding Protein 2 (IRP2), a protein that is degraded in iron-replete conditions [38]. We found that PIK-III treatment increased IRP2 protein levels (Fig 5B and 5C). Bioavailable iron can be complexed with transferrin and endocytosed by transferrin receptors (TFR) in mammalian cells [39]. To determine if VPS34 inhibition leads to reduced transferrin-mediated iron uptake, we monitored uptake of a fluorescently labelled transferrin probe. Endocytosis of transferrin probe was readily visible in vehicle treated cells, and pretreatment with PIK-III for 24 hours significantly impaired uptake of the probe (Fig 5D). Quantification revealed that PIK-III decreased the average transferrin probe intensity by 95% (Fig 5E). We next asked whether the transferrin receptor was impacted by VPS34 inhibition. Protein levels of TFR were strongly reduced by PIK-III as early as 1 hour in a dose-dependent manner (Fig 5F), and bafilomycin A1 or chloroquine prevented its clearance (Fig 5G and 5H). When VPS34 inhibition was prolonged to 72 hours, TFR protein levels were restored but not functional as transferrin probe uptake was still impaired (S2A Fig) and IRP2 levels not restored (S2B Fig). These data suggest that acute inhibition of VPS34 results in enhanced lysosomal delivery and degradation of the transferrin receptor, resulting in persistent cellular iron deprivation.

Besides TFR-mediated iron uptake, other mechanisms downstream of VPS34 could account for the iron starvation observed upon PIK-III treatment. Recent studies have shown that impaired endolysosomal acidification prevents lysosomal iron release depleting cells of iron [40, 41]. To rule out impaired acidification as a possible mechanism, we monitored the ratio of the pH-sensitive pHrodo Red versus the pH-insensitive Alexa Fluor 488 transferrin probes. As expected, the intensities of the transferrin probes reduced with 24 hours of PIK-III treatment but, the ratio of the probes did not change compared to vehicle treatment (S3A and S3B Fig) showing that PIK-III does not impact endosomal pH. Inhibition of acidification with bafilomycin A1 was used as a positive control and showed a decrease in the ratio of pHrodo Red versus Alexa Fluor 488 transferrin probe intensities (S3A and S3B Fig). Cells can also mobilize iron via the lysosome-mediated degradation of ferritin, a process known as ferritinophagy that relies on VPS34, NCOA4, TAX1BP1 and subset of autophagy components [1, 42, 43]. Based on project DRIVE data, RKO cells were highly sensitive to loss of TFR, but showed no dependency on ferritin, NCOA4 or TAX1BP1 (S4A Fig). In addition, we looked at the levels of ferritin and NCOA4 in PIK-III treated RKO cells. The low doses of PIK-III that resulted in clearance of TFR had minimal effects on NCOA4 and even decreased ferritin levels, which is a likely consequence of cellular iron depletion. Only higher concentration of PIK-III led to

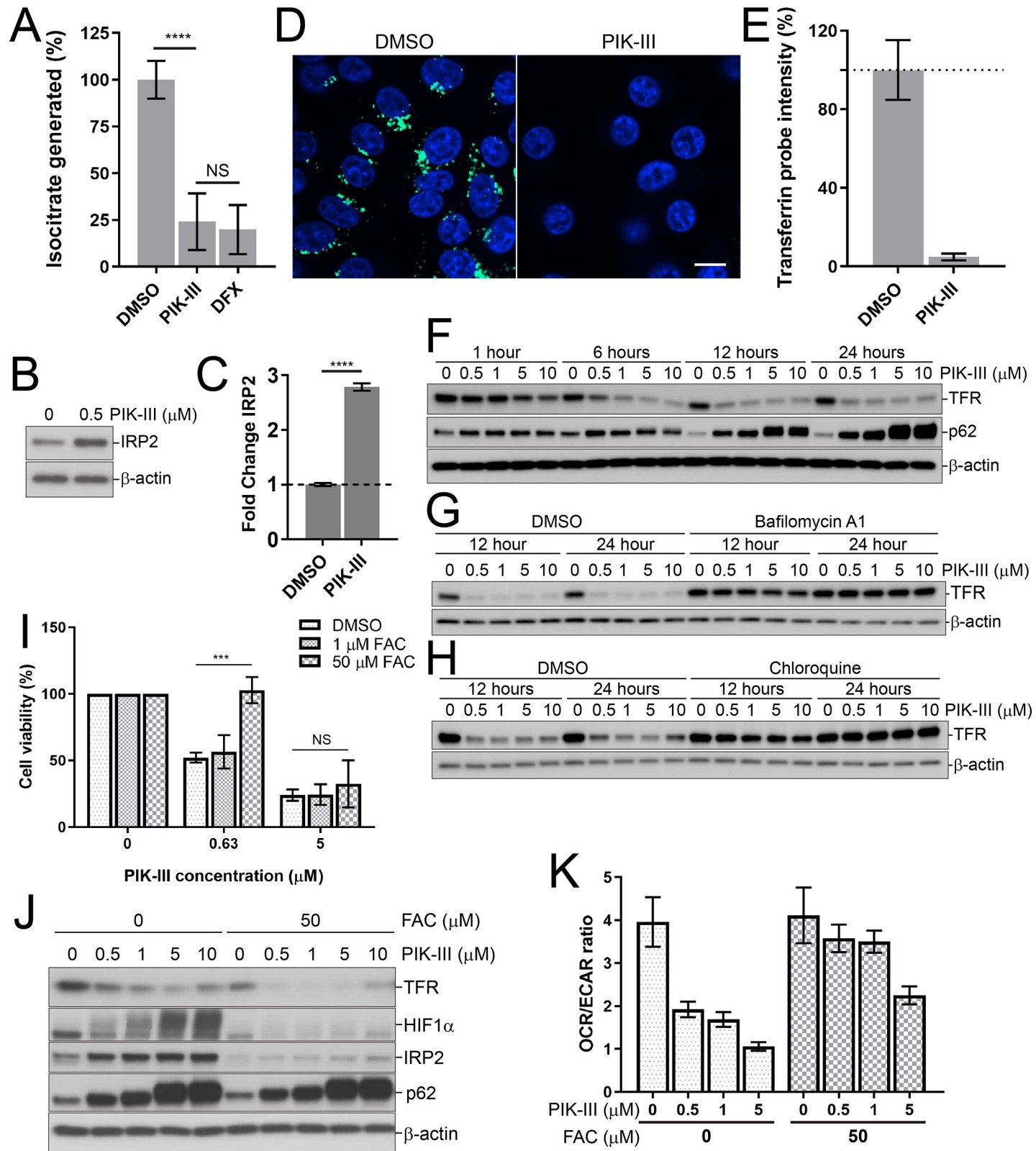

**Fig 5. VPS34 inhibition depletes intracellular iron by blocking iron uptake through enhanced degradation of TFR and excess iron rescues the growth defect in RKO cells.** (A) Aconitase activity in RKO cells upon VPS34 inhibition. RKO CTRL cells were treated with 1 µM PIK-III, 30 µM deferasirox (DFX) or vehicle control for 24 hours. The aconitase activity assay was used to quantify the isocitrate present in the cell extracts. All data was normalized to cells treated with vehicle, transformed to

represent the percent isocitrate generated and averaged from three biological replicates ± SD; ****, p < 0.0001; NS, not significant (two-way ANOVA). (B) RKO CTRL cells were treated with vehicle or 0.5 μM PIK-III for 24 hours and lysates were immunoblotted with the indicated antibodies. (C) Quantitation of immunoblot in (B) as fold change versus vehicle treated cells. All data were normalized to the β-actin loading control. Data is averaged from three independent experiments ± SD, ****, p < 0.0001 (two-way ANOVA). (D) RKO.Cas9 cells were treated with vehicle or 1 μM PIK-III for 24 hours followed by incubation with Transferrin Alexa Fluor 488 probe for 30 min at 37°C. Nuclei were labeled with Hoechst and live cell imaging was performed. The white bar represents 10 μm in length. (E) Cellular uptake of Transferrin Alexa Fluor 488 probe from panel (D) was quantified as percent average intensity of total cells. Data were averaged and presented as the mean ± SD from 6 wells. (F) RKO CTRL cells were treated with the indicated concentrations of PIK-III for 1, 6, 12, or 24 hours and lysates were immunoblotted with the indicated antibodies. (G, H) RKO CTRL cells were treated with the indicated concentrations of PIK-III with or without 50 nM Bafilomycin A1 (G) or 30 μM chloroquine (H) for 12 and 24 hours and lysates were immunoblotted with the indicated antibodies. (I) RKO CTRL cells were treated with indicated concentrations of PIK-III along with vehicle, 1 μM or 50 μM Ferric Ammonium Citrate (FAC) for 5 days and cell viability was assessed using the CellTiter-Glo assay. Data is derived from three independent experiments and presented as mean ± SD, ***, p < 0.001; NS, not significant (two-way ANOVA). (J) RKO CTRL cells were treated with the indicated concentrations of PIK-III along with vehicle or 50 μM FAC for 24 hours. Lysates were immunoblotted with the specified antibodies. (K) Mitochondrial respiration defect due to PIK-III treatment. RKO CTRL cells were treated with the indicated concentrations of PIK-III along with vehicle for 24 hours with or without 50 μM FAC and mitochondrial respiration was assessed by measuring the oxygen consumption rate (OCR) and extracellular acidification rate (ECAR) in response to the specified mitochondrial inhibitors using the Seahorse XFe96 analyzer. Oligomycin (Oligo) inhibits ATP synthase (Complex V) and decreases OCR; FCCP uncouples oxidative phosphorylation and increases OCR; and rotenone/antimycin A (ROT & AA) inhibits Complex I and III, respectively and decreases OCR. OCR and ECAR readouts from a single 96-well plate were normalized to the total DNA content measured in each well by DRAQ5 staining. Relative magnitude of mitochondrial oxidative phosphorylation and glycolysis is depicted as a ratio between the basal oxygen consumption rate and basal extracellular acidification rate (OCR/ECAR) and presented as the mean ± SD ($n$ = 11 or 12 wells). Individual OCR and ECAR graphs are shown in S5 Fig.

the accumulation of NCOA4 and ferritin, in line with blocked ferritinophagy (S4B Fig). These results indicate that low concentrations of PIK-III (0.5–1 μM) modulate TFR uptake, while higher concentrations of PIK-III (5–10 μM) are needed to block ferritinophagy.

## Iron deprivation underlies the VPS34-dependent growth defect and metabolic alterations in RKO cells

To evaluate the contribution of iron to the growth defect of RKO cells under VPS34 inhibition, we conducted rescue experiments with exogenous iron in the form of soluble ferric ammonium citrate (FAC). Excess iron was not only sufficient to rescue the growth defect of RKO cells treated with PIK-III at low concentrations (Fig 5I), but it also normalized HIF1α and IRP2 levels (Fig 5J). The addition of excess iron lowered TFR levels, as expected from the loss of IRP-mediated stabilization of TFR transcripts [44]. As a control, we analyzed p62, a well-characterized autophagy cargo receptor that is continuously turned over by autophagy [45]. Excess iron did not modulate PIK-III-mediated accumulation of p62, indicating that the defect in autophagy initiation remained unchanged (Fig 5J). Mitochondrial respiration is highly sensitive to iron deprivation as many enzymes of the mitochondrial oxidative phosphorylation system and tricarboxylic acid cycle use iron as a cofactor for electron shuttling [46, 47]. We assessed mitochondrial respiration by the Seahorse mitochondria stress test and found that PIK-III treatment reduced mitochondrial respiration by decreasing both basal and maximal oxygen consumption rates (OCR), while increasing extracellular acidification rates (ECAR) (S5A–S5D Fig). A decrease in the basal OCR/ECAR ratio after PIK-III treatment suggested that cells shift from oxidative phosphorylation towards glycolysis for energy production (Fig 5K). Importantly, the OCR/ECAR ratio after treatment with a low dose of PIK-III was normalized by the addition of excess soluble iron (Fig 5K). Therefore, we conclude that VPS34 activity can significantly influence metabolism and cell proliferation through the modulation of iron homeostasis, and iron deprivation drives the metabolic alterations and growth defect induced by VPS34 inhibition in RKO cells.

## Loss of RAB7A partially reverses iron deprivation, growth defect and metabolic alterations due to VPS34 inhibition

According to the CRISPR screen, the strongest genetic suppressor of PIK-III in RKO cells is RAB7A, a Ras-related protein that marks the late endosome compartment and is essential for

late endosome-lysosome fusion [48]. To substantiate our model that enhanced clearance of TFR drives iron starvation and cell growth defects upon PIK-III treatment, we asked the question whether RAB7A plays a role in TFR trafficking, iron deprivation, growth defect and metabolic alterations due to VPS34 inhibition. We first validated the knockout of *RAB7A* by immunoblot (Fig 6A) and confirmed that *RAB7A* KO rescued RKO colony formation in the presence of PIK-III (Fig 6B). Furthermore, *RAB7A* KO shifted the relative PIK-III IC50 ~9-fold from 0.28 ± 0.08 μM to 2.4 ± 0.4 μM in a cell viability assay (Fig 6C). Next, we asked if *RAB7A* KO reverses the enhanced lysosomal clearance of transferrin receptors. Loss of RAB7A not only restored TFR levels (Fig 6D), but also rescued the iron uptake defect as the transferrin probe was taken up in *RAB7A* KO cells treated with PIK-III (Fig 6E and 6F). Noteworthy, depletion of RAB7A alone resulted in an iron deprivation phenotype as transferrin probe uptake was basally reduced (Fig 6E and 6F) and IRP2 levels increased (Fig 6G). Consistent with the rescue of functional TFR and mobilization of iron, PIK-III treatment reduced IRP2 levels in *RAB7A* KO cells (Fig 6G). *RAB7A* KO also normalized HIF1α induction as well as OCR/ECAR ratio at low concentrations of PIK-III (Fig 6G and 6H). However, the iron starvation and mitochondrial respiration defects persisted at high concentrations of PIK-III, in line with deletion of RAB7A shifting the relative PIK-III IC50 in a cell viability assay (Fig 6C, 6G, 6H and S5E–S5H Fig). Based on these data, we propose that VPS34-RAB7A axis regulates iron homeostasis through the endolysosome-mediated iron acquisition pathway and this plays a key role in the growth of RKO cells.

## Discussion

This study identifies RKO as a highly VPS34-dependent cellular model and describes an iron-dependent metabolic vulnerability that underlies impaired RKO cell growth upon VPS34 inhibition. Our data suggests that regulation of endosomal maturation by VPS34 is essential for transferrin uptake, cellular iron availability, mitochondrial respiration, and ultimately, cell growth. Inhibition of VPS34 also impaired LDL uptake, but we found no evidence of cholesterol deprivation as key driver of the PIK-III sensitivity in RKO cells. In our attempt to reveal regulators of VPS34-dependence using a genome-wide CRISPR screen, *RAB7A* KO was identified as the top suppressor of the PIK-III-mediated growth defect. RAB7A defines the late endosome compartment and plays a critical role in the late endosome-lysosome fusion for cargo delivery to lysosomes for degradation [48]. Recently, VPS34 activity and PI(3)P-driven TBC1D2 recruitment has been shown to prevent RAB7A hyperactivation through a negative feedback loop [2]. We hypothesize that PIK-III drives RAB7A hyperactivation in RKO cells and deregulates trafficking and functionality of the transferrin receptor. Deletion of RAB7A may prevent transferrin receptors from physically entering the lysosomal compartment and promote recycling back to the cell surface [49]. In support of the endolysosomal blockade hypothesis, many of the other endolysosomal pathway components, which were identified as suppressors of the PIK-III-mediated growth defect, are directly involved in endosome maturation from RAB5-positive early endosomes to RAB7-positive late endosomes (Fig 2C–2E). One such example is the CCZ1-MON1 complex, which acts as a guanine nucleotide exchange factor (GEF) and is responsible for recruiting and activating RAB7A [50, 51]. The CCZ1-MON1 complex is first recruited by activated RAB5 present on early endosomes with the help of the class C core vacuole/endosome tethering (CORVET) complex [50]. After RAB7A recruitment and activation, the CCZ1-MON1 complex assists in the removal of RAB5 (Fig 2E). The PIK-III suppressor RMC1 was recently identified as a component of the CCZ1-MON1 complex and acts as positive regulator to help recruit RAB7 to late endosomes and autophagosomes [52]. Similar to the depletion of RAB7A, we suggest that loss of the CCZ1-MON1-RMC1 complex

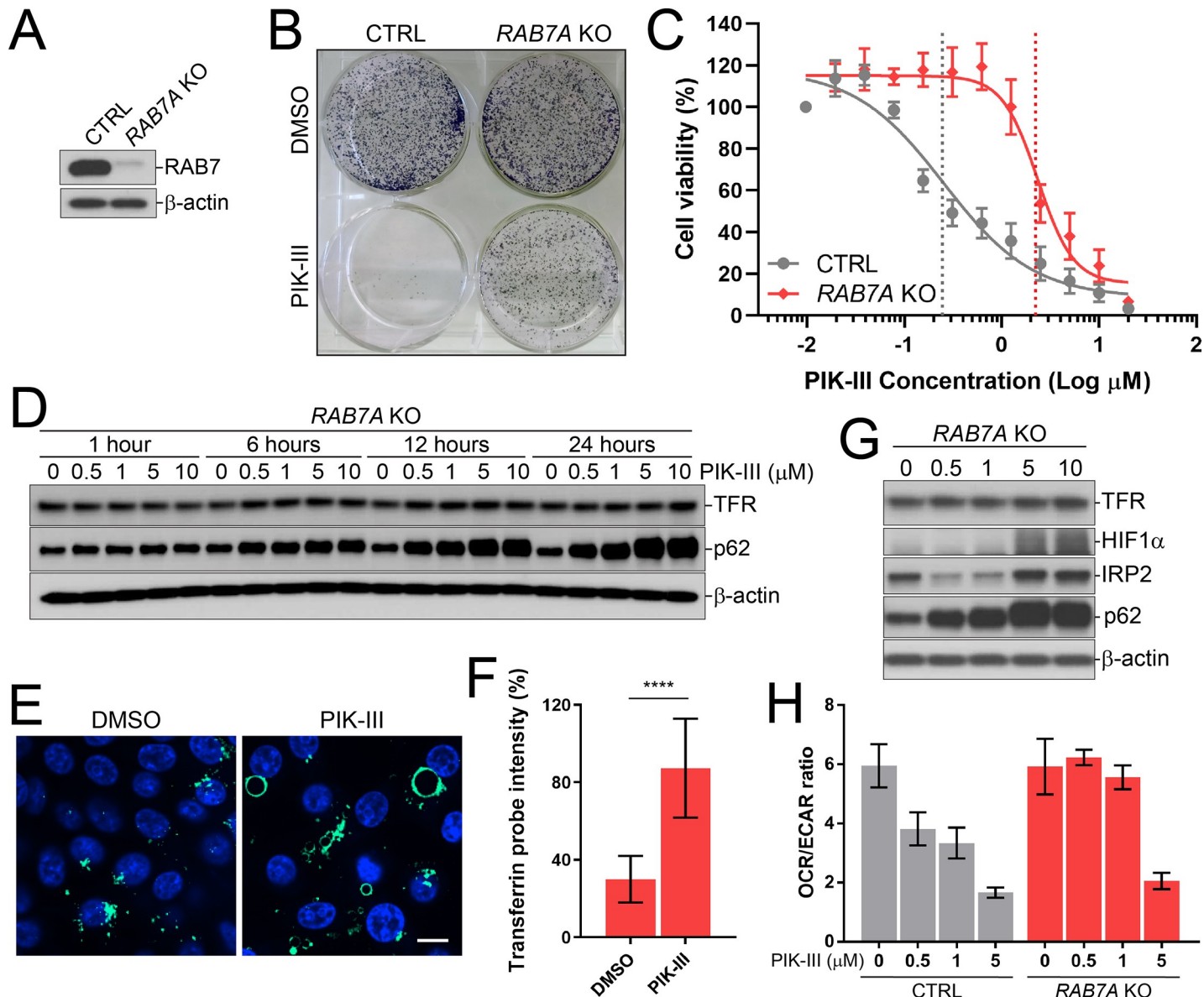

**Fig 6. *RAB7A* KO partially rescues PIK-III dependency of RKO cells.** (A) Characterization of *RAB7A* KO cells. Cell lysates of RKO CTRL and *RAB7A* KO cells were immunoblotted with the specified antibodies. (B) Knockout of RAB7A suppresses VPS34 vulnerability. RKO CTRL and *RAB7A* KO cells were treated with 1 μM PIK-III or vehicle control for 9 days and cell colonies were visualized with crystal violet staining. (C) RKO CTRL and *RAB7A* KO cells were treated with PIK-III in dose response for 5 days and cell viability was assessed using the CellTiter-Glo assay. The dose response was collected from three independent experiments and represented as the mean ± SD. Non-linear regression analysis with variable slope and least squares fit was performed using GraphPad Prism 8 to calculate the IC50 values. The IC50 inflection points for RKO CTRL and *RAB7A* KO are visualized by a dotted line and colored dark grey and red, respectively. (D) *RAB7A* KO cells were treated with the indicated concentrations of PIK-III for 1, 6, 12, or 24 hours and lysates were immunoblotted with the specified antibodies. (E) RKO *RAB7A* KO cells were treated with vehicle or 1 μM PIK-III for 24 hours followed by incubation with Transferrin Alexa Fluor 488 probe for 30 min at 37˚C. Nuclei were labeled with Hoechst and live cell imaging was performed. The white bar represents 10 μm in length. (F) Transferrin Alexa Fluor 488 probe uptake in (E) was quantified as percent average intensity of total cells. Data in the form of technical replicates were averaged and presented as the mean ± SD from 6 wells. (G) *RAB7A* KO cells were treated with the indicated concentrations of PIK-III for 24 hours. Lysates were immunoblotted with the specified antibodies. (H) Mitochondrial respiration defect due to VPS34 inhibition is RAB7A-dependent. RKO CTRL or *RAB7A* KO cells were treated with the indicated concentrations of PIK-III for 24 hours and mitochondrial respiration was assessed by measuring OCR and ECR as described in Fig 5K. The OCR/ECAR ratio is presented as the mean ± SD (*n* = 7 or 8 wells). Individual OCR and ECAR graphs are shown in S5 Fig.

impairs delivery of transferrin receptors to the lysosome, thereby preventing the degradation of TFR in the context of VPS34 inhibition.

There exists conflicting reports on the role of VPS34 in receptor trafficking. Chen *et al*, observed that wortmannin and LY294002 enhanced the lysosomal degradation of EGF-induced EGFR degradation [53]. On the other hand, wortmannin has been shown to disrupt the trafficking of many receptors and transporters such as EGFR, β2AR, GLUT1, PDGFR and TFR through the disruption of RAB5 function at early endosomes [7, 54–56]. Furthermore, genetic knockout of VPS34 was shown to block the lysosomal degradation of EGFR, but not transferrin receptor [2]. Paradoxically, in RKO cells, we observed accelerated clearance of TFR upon VPS34 inhibition by PIK-III, and clearance occurred by lysosomal degradation since TFR levels were restored by bafilomycin A1 and chloroquine treatment (Fig 5G and 5H). Increased clearance of TFR upon VPS34 inhibition appears to be exceptionally strong in RKO since only modestly reduced TFR levels were observed in other cell lines such as H4, DLD1 or KYSE70 (S6 Fig). Overall, our data suggests that RKO cells may represent a sensitized system to reveal the transient trafficking effects downstream of VPS34.

While VPS34 activity regulates many signaling and trafficking pathways, excess exogenous iron was sufficient to rescue the PIK-III-mediated growth defect in RKO cells at low PIK-III concentrations. In this setting, we propose that soluble iron enters the cell through transferrin-independent mechanisms, likely through low-affinity divalent metal ion transporters. Two SLC39 family members, ZIP8 and ZIP14, have been identified to transport ferrous iron across the cell membrane [57]. Additionally, the lysosomal iron transporter SLC11A2 possesses a cell membrane isoform (SLC11A2.2), which is capable of transporting iron from the extracellular environment [41]. The rescue of cell proliferation by excess exogenous iron implies that iron deprivation underlies VPS34-dependency in RKO cells. Recent publications describe that impaired lysosomal acidification triggers cellular iron deficiency, leading to impaired mitochondrial respiration and cell proliferation with supplementation of excess iron being sufficient to restore the iron deficiency responses [40, 41]. However, our results show that acidification of transferrin-containing endosomal compartments is unchanged upon PIK-III treatment and unlikely responsible for the iron deficiency effects. Iron is a nutrient used in many important biological processes due to its redox-active properties. Cellular iron deprivation negatively impacts many iron-dependent metabolic pathways such as the tricarboxylic acid cycle and the electron transport chain [33]. Iron deprivation induced by iron chelation is also known to inhibit cell growth by reducing ribonucleotide reductase activity and lower the levels of deoxyribonucleotides necessary for DNA synthesis in S-phase [35]. Iron levels are tightly regulated at the translational level by iron responsive proteins (IRP1/2) that recognize iron-responsive elements (IRE) present in the 5' or 3' untranslated regions of iron metabolism-associated mRNA [58]. Iron deprivation triggers TFR mRNA stabilization and thereby allows the production of more TFR to increase iron uptake through the endolysosomal pathway [44]. Concurrently, iron stored in ferritin is released through a process called ferritinophagy [1, 42, 43]. Ferritinophagy requires VPS34 activity as inhibition with PIK-III completely blocks ferritin degradation under iron deprivation conditions [1]. Thus, VPS34 plays a dual role in regulating iron homeostasis by controlling TFR recycling and iron uptake at early RAB5-positive endosomes as well as ferritin acquisition and degradation at late RAB7-positive endosomal/lysosomal compartments. This study suggests a potential VPS34 dosage effect on these two iron mobilization pathways. While low concentrations of PIK-III (0.5–1 μM) modulate TFR uptake, higher concentrations of PIK-III (5–10 μM) are needed to block ferritinophagy (S4B Fig). This proposed model deserves further investigation and validation with other PIK-III sensitive cell lines in future studies.

Interestingly, a role for VPS34 in iron homeostasis has also been described in the yeast *Candida glabrata* [59]. In *C. glabrata*, CgVPS34 regulates iron homeostasis by controlling the

retrograde trafficking of CgFtr1, a high-affinity iron transporter, through the endosomal pathway. Loss of CgVPS34 prevents shuttling of CgFtr1 from the cell surface to the central vacuole in response to high environmental iron levels and results in the over-accumulation of intracellular iron, leading to a growth defect [59]. While the predominant iron import systems and trafficking routes differ between yeast and mammalian cells, the utilization of VPS34 as a regulator of iron homeostasis could possibly be evolutionarily conserved.

Lastly, some of the cellular consequences to VPS34 inhibition we have described in RKO cells have been observed in mouse models. *Zhou et al.*, have shown that VPS34 mutant mouse embryos display a reduced cell proliferation rate that is not attributed to programmed cell death [10]. Bilanges *et al.*, have shown that heterozygous inactivation of VPS34 in mice decreases mitochondrial respiration without negatively affecting mitochondrial integrity [8]. In light of our data, it will be interesting to evaluate if partial inactivation of VPS34 may induce cellular iron deprivation at the organismal level, contributing to decreased cell proliferation rates and decreased mitochondrial respiration.

## Materials and methods

### Antibodies, chemicals and reagents

The following primary antibodies were used: Rabbit anti-VPS34 (Cell Signaling Technologies, 4263S; 1:1000 WB), mouse anti-α-tubulin (Cell Signaling Technologies, 2144; 1:1000 WB), rabbit anti-TetR (Novus Biologicals, NB600-234; 1:1000 WB), rabbit anti-ATG5 (Cell Signaling Technologies, 12994S; 1:1000 WB), rabbit anti-ATG7 (Cell Signaling Technologies, 8558S; 1:1000 WB), rabbit anti-ATG16L1 (Cell Signaling Technologies, 8089S; 1:1000 WB), mouse anti-SREBP2 (BD Biosciences, 557037; 1:1000 WB), rabbit anti-HIF1α (Cell Signaling Technologies, 14179S; 1:1000 WB), mouse anti-β-actin (Cell Signaling Technologies 3700S, 1:5000 WB), rabbit anti-IRP2 (Cell Signaling Technologies, 37135S; 1:1000 WB), mouse anti-p62 (BD Biosciences, 610833; 1:3000 WB), rabbit anti-LDLR (Abcam, ab52818; 1:1000 WB), rabbit anti-TFR (Cell Signaling Technologies, 13113S; 1:1000 WB), rabbit anti-FTH1 (Cell Signaling Technologies, 4393; 1:1000 WB), mouse anti-NCOA4 (Santa Cruz Biotechnology, sc-373739; 1:1000 WB), goat anti-CTSL (R&D Systems, AF952; 1:1000 WB) and rabbit anti-Rab7 (Cell Signaling Technologies, 9367S; 1:1000 WB). The following secondary antibodies were used: Goat anti-mouse IgG HRP (Cell Signaling Technologies, 7076S; 1:5000 WB), goat anti-rabbit IgG HRP (Cell Signaling Technologies, 7074S; 1:5000 WB) and Donkey anti-goat IgG HRP (ThermoFisher Scientific, A15999; 1:5000 WB). For DNA staining: Hoechst 33342 (Thermo Fisher Scientific, H3570; 1:1000 IF).

The following chemicals were used: PIK-III (Novartis Pharmaceuticals), deferasirox (DFX) (Novartis Pharmaceuticals), bafilomycin A1 (Sigma Aldrich), chloroquine (Sigma Aldrich), ferric ammonium citrate (Sigma Aldrich), Transferrin Alexa Fluor 488 (Thermo Fisher Scientific), pHrodo Red Transferrin (Thermo Fisher Scientific), Bodipy FL LDL (Thermo Fisher Scientific), cholesterol-water soluble (Sigma Aldrich), and puromycin (VWR), blasticidin (Thermo Scientific), neomycin (Thermo Scientific) and doxycycline (DOX) (Sigma Aldrich).

The following vectors were used: pLKO-Tet-On for shRNA knockdown [60, 61], pLenti4/TO/V5-DEST for protein overexpression (ThermoFisher Scientific), pNGx-LV-c004 for Cas9 expression and pNGx-LV-g003 for sgRNA CRISPR knockout [62].

### Mammalian cell culture

Cancer cell lines used in this study were obtained from the Novartis Cell Databank. RKO cells were cultured in high glucose Dulbecco's modified Eagle's medium (DMEM) supplemented with 10% fetal bovine serum at 37˚C and 5% $CO_2$ in a humidified incubator. RKO, RKO.Cas9,

RKO.Cas9 CTRL (RKO CTRL) and RKO.Cas9 *RAB7A* KO (RKO *RAB7A* KO) cells were cultured in high glucose, no glutamine Dulbecco's modified Eagle's medium (DMEM) supplemented with 2 mM glutamine and 10% fetal bovine serum at 37˚C and 5% $CO_2$ in a humidified incubator. H4 and DLD1 cells were also cultured in high glucose, no glutamine Dulbecco's modified Eagle's medium (DMEM) supplemented with 2 mM glutamine and 10% fetal bovine serum at 37˚C and 5% $CO_2$ in a humidified incubator. KYSE70 cells were grown in Roswell Park Memorial Institute (RPMI) 1640, 10% fetal bovine serum supplemented with 2 mM glutamine at 37˚C and 5% $CO_2$ in a humidified incubator. All cell culture reagents were obtained from Invitrogen.

Inducible shRNA plasmids were generated using shRNA sequences listed in Table 1 and cloned into the doxycycline-inducible pLKO-Tet-On vector [60, 61]. The stable dox inducible shRNA RKO cell lines were generated by lentiviral delivery of pLKO-Tet-On-shRNA constructs and selected with puromycin. VPS34-overexpression cell lines were generated by lentiviral delivery of shRNA-resistant pLenti4/TO/V5-DEST-VPS34-WT (wild type), or pLenti4/TO/V5-DEST-VPS34-KD (K636R) and selected with neomycin. The RKO-Cas9 cells were generated by lentiviral delivery of Cas9 in pNGx-LV-c004 and selected with blasticidin. sgRNA for RAB7A was selected based on the CRISPR screen results and cloned into the BbsI restriction site of pNGx-LV-g003 lentiviral backbone as previously described [42]. A scramble sgRNA was used to generate the RKO CTRL. The sequences for all individual sgRNA are listed in Table 1.

For lentiviral transduction, $8 \times 10^5$ HEK293T cells were plated in a 6-well plate and transfected next day with lentiviral packaging mix (1 μg ΔVPR (Δ8.9) and 0.25 μg VSV-G) along with1 μg of lentiviral vector using FuGene 6 (Promega). Viral particle-containing supernatants were collected after 48 hours, centrifuged at 1500 rpm for 5 minutes and syringe filtered using a 0.45 μm filter (Millipore). Polybrene was added to a final concentration of 10 μg/mL and target cells were infected overnight. Next day, media was aspirated and cells were replenished with fresh media and allowed to recover for 24 hours. Cells were then selected with 2 μg/mL puromycin for 72 hours. Transduced and selected CRSIPR KO cell lines were used as total cell populations.

## Cell line profiling with PIK-III

Cell lines ($n = 306$) were maintained in humidified incubators at 37˚C and 5% $CO_2$ in DMEM or RPMI 1640 supplemented with 10% FBS. Cells were seeded into 1,536-well plates, were allowed to adhere for 12–24 h, and then were treated with an 8-point concentration-response of PIK-III with a maximal final concentration of 8 μM. Each compound concentration was tested in duplicate. After 72 h, cell proliferation was assessed relative to 0.4% DMSO (0%) and 1 μM MG132 (100%) using CellTiter-Glo (Promega), and IC50 values were determined from a curve fitted to duplicate 8-point concentration-response datasets from two independent assay plates. Data is presented in S1 Table.

**Table 1. sgRNA and shRNA sequences.**

| Gene Name | sgRNA | Sequence |
|---|---|---|
| CTRL (scramble) | 1 | GTAGCGAACGTGTCCGGCGT |
| *RAB7A* | 1 | ACGGTTCCAGTCTCTCGGTG |
| **Gene Name** | **shRNA** | **Sequence** |
| *VPS34* | 2486 | CCGGCGAAGGTATTCTAATCTGATTCTCGAGAATCAGATTAGAATACCTTCGTTTTTG |
| *ATG5* | 915 | CCGGCCTTTCATTCAGAAGCTGTTTCTCGAGAAACAGCTTCTGAATGAAAGGTTTTTG |
| *ATG7* | 1520 | CCGGGCTTTGGGATTTGACACATTTCTCGAGAAATGTGTCAAATCCCAAAGCTTTTT |
| *ATG16L1* | 2181 | CCGGGCCTGGAAGAATAACACTGAACTCGAGTTCAGTGTTATTCTTCCAGGCTTTTTG |

## Cell line profiling by pooled shRNA screening

Data was extracted from Project DRIVE [25] and RSA values are visualized.

## Colony formation assay

For the VPS34 shRNA experiment, 1000 cells were plated per well in 6-well tissue culture (TC) treated dishes and after overnight incubation, the medium was changed to DMEM containing either 10% (high) or 0.5% (low) fetal bovine serum with or without 10 ng/mL doxycycline. The medium was replenished every 3 days for 9 days. For the PIK-III treatment experiment, RKO CTRL or *RAB7A* KO cells were plated at 10,000 cells per well in 6-well TC-treated dishes and after 24 hours, 0.1% DMSO or 0.5 μM PIK-III was added. The medium was replenished every 3 days for 8 day. Plates were then washed twice with PBS, air dried, and stained with crystal violet solution (0.2%; Boston BioProducts) for 20 minutes at room temperature. After staining, plates were again washed 4 times with deionized water and air dried for at least 16 hours at room temperature before imaging with a conventional camera.

## Viability assays

For the shRNA experiments, 1000 cells were plated per well in 96-well TC-treated dishes and after overnight incubation, the medium was changed to DMEM containing either 10% (high) or 0.5% (low) fetal bovine serum with or without 10 ng/mL doxycycline. Plates were read out at days 0, 3, 5 and 7 to assess cell viability by using the CellTiter-Glo assay (Promega; G7571) according to manufacturer's instructions. RKO ATG5, ATG7 and ATG16L1 shRNA cell lines were grown in 0.5% fetal bovine serum with or without 10 ng/mL doxycycline and plates were read out at days 3, 5, 7 and 9 to assess cell viability by using the CellTiter-Glo assay (Promega; G7571) according to manufacturer's instructions.

For the PIK-III dose response experiment, RKO CTRL or *RAB7A* KO cells were plated at 2000 cells per well in 96-well white TC-treated dishes (Corning; 3610). The following day, PIK-III was added in 10 pt 2-fold dilution (20 μM to 9.8 nM) and after 5 days of incubation, cell viability was assessed using CellTiter-Glo assay (Promega; G7571) according to manufacturer's instructions. IC50 values were calculated in GraphPad Prism 8 using non-linear regression with variable slope and least squares fit.

For the cholesterol and iron cell proliferation rescue experiments, RKO cells were plated at 2000 cells per well in 96-well white TC-treated dishes (Corning; 3610) in media containing FAC or soluble cholesterol at their indicated concentrations. The following day, PIK-III was added at 0, 0.63 μM and 5 μM concentrations and after 5 days of incubation, cell viability was assessed using CellTiter-Glo assay (Promega; G7571) according to manufacturer's instructions. Statistics were calculated using GraphPad Prism 8 using two-way ANOVA.

## Genome-wide CRISPR screen and validation

A genome-wide sgRNA library targeting 18,360 genes with 5 sgRNAs per gene was constructed as previously described [62]. Packaging of the lentiviral sgRNA library was done according to DeJesus *et al.* [62]. RKO-Cas9 cells were transduced with the lentiviral sgRNA library and after 14 days, the cells were treated with 0.1% DMSO or 0.5 μM PIK-III. The cells were then incubated for 28 days with PIK-III-supplemented medium changes every 3 days. Cells were split as required and seeded back to 55 million cells per 5-stack cell culture chamber (Corning). Genomic DNA from the live cells was isolated followed by sequencing as previously described [62].

## Gene expression profiling with RNA-seq

RKO cells at 1 million cells per well were plated in 6-well tissue culture plates and the next day were treated with 0.1% DMSO or 0.5 μM PIK-III for 24 hours. RNA was extracted using the RNeasy Plus Kit (Qiagen) according to the manufacturer's protocol. RNA sequencing libraries were prepared using the Illumina TruSeq Stranded mRNA Sample Preparation protocol, sequenced using the Illumina HiSeq4000 platform and analyzed as described previously [62]. The following differential tests were performed: RKO_DMSO versus RKO_PIK-III. Results were reported in terms of log2 fold changes and negative log10 adjusted P values (S3 Table) (Benjamini Hochberg false discovery rate). Gene set enrichment analysis was performed using the Molecular Signatures Database hallmark gene sets [26–28]. RNA-seq raw sequencing data have been deposited in the Sequence Read Archive under accession number PRJNA633293.

## Fluorescent probe assays

RKO.Cas9 or *RAB7A* KO cells were seeded at 3000 cells per well in 384 well microplates (Perkin Elmer CellCarrier). The cells were treated with 0.1% DMSO or 1 μM PIK-III for 24 hours. Transferrin Alexa Fluor 488, Bodipy FL LDL and Hoechst 33342 were added at 5 ug/mL, 25 ug/mL and 10 μM respectively. The LDL probe was incubated for 2 hours at 37˚C while the transferrin probe was incubated for 30 minutes at 37˚C. Incubation times were adjusted to accommodate differences in probe intensities upon accumulation in cells. Cells were imaged live at 60× objective magnification on a CV7000 automated confocal microscope (Yokogawa). Nuclei were imaged on the blue channel (405 nm excitation), and chemical fluorescent probes were imaged on the green channel (488 nm excitation) and 8 fields were collected per well. Images were quantified using Yokogawa analysis software. Briefly, nuclei were identified using dynamic threshold module and non-specific cytoplasmic Hoechst staining identified the cell body. The images were then masked to identify chemical fluorescent probe objects in the green channel and total cell mean intensities were measured from within the defined cell parameters and averaged per well.

To assess endosomal acidification, RKO.Cas9 cells were seeded between 5000 to 15000 cells per well in 96 well microplates (GreinerOne). Cells were treated with 0.1% DMSO or 1 μM PIK-III for 24 or 72 hours, or bafilomycin A1 at 100 nM for 4 hours. Transferrin Alexa Fluor 488 and pHrodo Red Transferrin were added each to a final concentration of 5 ug/mL in a 1:1 mixture. The transferrin probes were incubated for 20 minutes at 37˚C followed by a media change before imaging. Cells were imaged live at 40× objective magnification on an IN Cell Analyzer 6000 (GE Healthcare). Chemical fluorescent probes were imaged on the green channel (488 nm excitation) and red channel (561 nm excitation) and 12 fields were collected per well. We developed an image analysis workflow using the Cell Profiler open-source platform [63]. Briefly, the analysis used the Alexa Fluor 488 channel to determine the image regions that are transferrin-positive. We then determined the median intensity within this region for both the Alexa Fluor 488 and the pHrodo Red channels. The ratio of these two measurements were taken as the readout to quantify acidification, normalized against DMSO treatment.

## Immunoblotting

Cells were lysed in RIPA buffer (Cell Signaling Technology) supplemented with 1% sodium dodecyl sulfate (SDS, Boston BioProducts), protease inhibitors (cOmplete EDTA-free, Roche) and 100 nM Calyculin A (Cell Signaling Technology). Lysates were homogenized using Qiashredder columns (QIAGEN) and protein concentration was determined using Lowry DC assay (Bio-Rad). The solubilized lysate was denatured in 6× Laemmli SDS loading buffer (Boston BioProducts) at 100˚C for 8 minutes. Samples with equal protein concentration were separated

using a Criterion$^{TM}$ TGX$^{TM}$ 4–15% gel system (Bio-Rad); transferred to Trans-Blot Turbo, 0.2 μm nitrocellulose membrane (Bio-Rad); and subjected to immunoblotting using standard methods.

## Aconitase activity assay

RKO cells were plated at 2 million cells in 6 cm plates and treated with 0.1% DMSO, 1 μM PIK-III or 30 μM DFX for 24 hours. Cells were harvested, washed twice with cold PBS and resuspended in assay buffer. Cells were then sonicated and centrifuged to remove insoluble material. Cell lysates were normalized using the DC protein assay (Bio-Rad) to a final total protein concentration of 3.6 mg/ml. The aconitase activity assay (Sigma Aldrich; MAK051) was run according to the manufacturer's instructions except that the assay incubation was for 120 minutes at 25˚C before addition of the developer. A standard curve derived from known concentrations of isocitrate was used to calculate the isocitrate generated from each unknown sample and presented in nmole units.

## Lipidomics

RKO cells were plated in 10 cm plates and treated with 0.1% DMSO or 1 μM PIK-III for 24 hours. Three biological replicates were collected for each condition. Lipid extracts were prepared using a modified methyl tert-butyl ether (MTBE) extraction protocol as previously described [64]. Briefly, adherent cells were harvested, resuspended and homogenized in a solution of methanol:water (1.5:1) and lipids were extracted with MTBE using the protocol established by Breitkopf and colleagues [64]. Extracted lipids were dried under vacuum and stored at -20˚C. Extracts were analyzed by LC-MS by the Mass Spectrometry, Metabolomics, Lipidomics and Proteomics Core group at Beth Israel Deaconess Medical Center. For data analysis, only lipid ion species that were detected in all experimental replicates were analyzed. Relative abundance measurements from LC-MS were log2 transformed for parametric statistical analysis (ANOVA) on aggregated lipid classes. Lipid classes were considered significantly changed if adjusted P-value (Benjamini-Hochberg procedure) was less than 0.05.

## Seahorse XFe96 mitochondria stress test assay

Oxygen consumption and extracellular acidification rates were measured using the Seahorse XF Cell Mito Stress Test Kit (Agilent) on the Seahorse Bioscience XFe96 analyzer (Agilent) according to the manufacturer's protocols. Briefly, 25,000–30,000 RKO CTRL or *RAB7A* KO cells were plated in XF96 cell culture microplates and treated with 0.1% DMSO and PIK-III for 24 hours. For some conditions, cells were also co-treated with 50 μM ferric ammonium citrate. Plates were washed with non-buffered DMEM containing 1 mM pyruvate, 2 mM glutamine and 25 mM glucose with pH adjusted to 7.4. Baseline oxygen consumption rates (OCR) and extracellular acidification rates (ECAR) were measured three times before sequential injection of oligomycin (final concentration 1 μM), FCCP (final concentration 0.5 μM) and a mixture of rotenone (final concentration 0.5 μM) and antimycin A (final concentration 0.5 μM) to measure uncoupled respiration, spare respiration capacity, and non-mitochondrial respiration, respectively. All OCR and ECAR measurements were normalized to DNA content with DRAQ5 staining (Thermo Scientific) using the Seahorse Wave Software (Agilent). The basal OCR/ECAR ratio was calculated by dividing the final basal time point OCR against its corresponding basal ECAR value from each well and then the OCR/ECAR ratios from each treatment group were averaged.

## Supporting information

**S1 Fig. The autophagy pathway is not involved in the RKO cell proliferation defect.** (A) Pooled shRNA screening data for VPS34, ATG5, ATG7, and ATG16L1 was extracted from project DRIVE [25] and visualized as RSA significance scores. Each dot represents a cell line and RKO is highlighted in red. The straight and dotted lines indicate average and 2x standard deviation of the RSA values across all cell lines, respectively. (B) Knockdown of canonical autophagy components (ATG5, ATG7 and ATG16L1) were performed in RKO cells expressing doxycycline (Dox) inducible shRNA targeting ATG5, ATG7 or ATG16L1. Cell proliferation was monitored by CellTiter-Glo assay as the mean of three independent experiments with error bars representing SD. *, $p < 0.05$; **, $p < 0.01$; ***, $p < 0.001$ (two-way ANOVA). (C) Characterization of RKO ATG5, ATG7 and ATG16L1 doxycycline (Dox) inducible shRNA cell lines. Cell lysates of RKO ATG5, ATG7 and ATG16L1 cells were immunoblotted with the specified antibodies.
(TIF)

**S2 Fig. Long-term VPS34 inhibition continues to block transferrin uptake.** (A) RKO CTRL cells were treated with vehicle or 1 μM PIK-III for 72 hours followed by incubation with Transferrin Alexa Fluor 488 probe for 20 min at 37˚C and live cell imaging was performed. The white bar represents 10 μm in length. (B) RKO CTRL cells were treated with the indicated concentrations of PIK-III for 12, 24 and 72 hours and lysates were immunoblotted with the indicated antibodies.
(TIF)

**S3 Fig. VPS34 inhibition does not impact endosomal pH.** (A) RKO CTRL cells were treated with vehicle or 1 μM PIK-III for 24 hours and with 100 nM bafilomycin (BAFA1) for 4 hours. The cells were incubated with 1:1 mixture of the Alexa Fluor 488 and pHrodo Red transferrin probes for 20 min at 37˚C before live cell imaging. The white bar represents 10 μm in length. (B) Images collected in (A) were quantified as the percent ratio of median intensity of pHrodo Red and Alexa Fluor 488 transferrin probes. Data was averaged and presented as the mean ± SD from 2 wells.
(TIF)

**S4 Fig. Characterization of ferritinophagy in RKO cells.** (A) Pooled shRNA screening data for VPS34, TFR, FTH1, NCOA4 and TAX1BP1 were extracted from project DRIVE [25] and visualized as RSA significance scores. Each dot represents a cell line and RKO is highlighted in red. The straight and dotted lines indicate average and 2x standard deviation of the RSA values across all cell lines. (B) Cell lysates of RKO CTRL cells treated with the indicated concentrations of PIK-III for 24 hours were immunoblotted with the specified antibodies.
(TIF)

**S5 Fig. Excess soluble iron or loss of RAB7A restores RKO metabolic homeostasis under VPS34 inhibition.** (A–D) Mitochondrial respiration defect due to PIK-III treatment. RKO CTRL cells were treated with the indicated concentrations of PIK-III along with vehicle for 24 hours and mitochondrial respiration rates for OCR, (A) and ECAR, (B) were assessed using the Seahorse XFe96 analyzer. Data in the form of technical replicates were averaged and presented as the mean ± SD ($n$ = 11 or 12 wells). RKO CTRL cells were treated with the indicated concentrations of PIK-III along with 50 μM FAC for 24 hours and mitochondrial respiration was assessed by measuring the OCR, (C) and the ECAR, (D). Data in the form of technical replicates were averaged and presented as the mean ± SD ($n$ = 11 or 12 wells). (E–H) Mitochondrial respiration defect due to VPS34 inhibition is RAB7A-dependent. RKO CTRL or *RAB7A*

KO cells were treated with the indicated concentrations of PIK-III for 24 hours and mitochondrial respiration was assessed by measuring the OCR (E, G) or the ECAR (F, H). Data in the form of technical replicates were averaged and presented as the mean ± SD ($n$ = 7 or 8 wells). (TIF)

**S6 Fig. VPS34 inhibition promotes lysosomal degradation of transferrin receptor in other cells lines.** H4, DLD1 and KYSE70 cells were treated with the indicated concentrations of PIK-III for 24 hours and lysates were immunoblotted with the indicated antibodies. (TIF)

**S1 Table. Compound cell line profiling.** List of IC50 and maximal activity values across cell lines treated with PIK-III. (XLSX)

**S2 Table. Genome-wide pooled CRISPR screen.** Gene-level data for the proliferation-based pooled CRISPR screen. (XLSX)

**S3 Table. Transcriptomic profiling.** RNA-seq data collected for RKO cells treated with PIK-III. (XLSX)

**S4 Table. Lipidomics profiling.** Lipid metabolite analysis for RKO cells treated with PIK-III. (XLSX)

**S1 Raw images.** (TIF)

## Acknowledgments

We would like to thank Salil Srivastava for training on the Seahorse XFe96 analyzer, Mark-Anthony Bray for help with image analysis, and Felipa Mapa, Howard Miller, Jessi Ambrose, Peter Aspesi, and Nathan Ross for performing the PIK-III cell line profiling.

## Author Contributions

**Conceptualization:** Marek J. Kobylarz, Judith Knehr, Leon O. Murphy, Suchithra Menon, Beat Nyfeler.

**Data curation:** Marek J. Kobylarz, Jonathan M. Goodwin, Zhao B. Kang, John W. Annand, Sarah Hevi, Ellen O'Mahony, John Reece-Hoyes, Alicia Lindeman, Martin Beibel, Guglielmo Roma, Walter Carbone, Judith Knehr, Dmitri Wiederschain.

**Formal analysis:** Marek J. Kobylarz, Zhao B. Kang, John W. Annand, Ellen O'Mahony, Gregory McAllister, Carsten Russ, Alicia Lindeman, Martin Beibel, Guglielmo Roma, Walter Carbone, Judith Knehr, Joseph Loureiro, Christophe Antczak, Dmitri Wiederschain.

**Investigation:** Marek J. Kobylarz, Jonathan M. Goodwin, Dmitri Wiederschain.

**Methodology:** Marek J. Kobylarz, Jonathan M. Goodwin, Sarah Hevi, Ellen O'Mahony, John Reece-Hoyes, Qiong Wang, John Alford, Joseph Loureiro, Christophe Antczak, Dmitri Wiederschain.

**Project administration:** Marek J. Kobylarz.

**Resources:** John W. Annand, John Reece-Hoyes, Qiong Wang, John Alford, Joseph Loureiro, Christophe Antczak, Leon O. Murphy, Suchithra Menon, Beat Nyfeler.

**Software:** Zhao B. Kang, Gregory McAllister, Joseph Loureiro, Christophe Antczak.

**Supervision:** Leon O. Murphy, Suchithra Menon, Beat Nyfeler.

**Validation:** Marek J. Kobylarz.

**Visualization:** Marek J. Kobylarz, Gregory McAllister, Martin Beibel, Guglielmo Roma, Walter Carbone, Judith Knehr, Dmitri Wiederschain.

**Writing – original draft:** Marek J. Kobylarz.

**Writing – review & editing:** Marek J. Kobylarz, Jonathan M. Goodwin, Leon O. Murphy, Suchithra Menon, Beat Nyfeler.

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
