## [Decision Letter · Decision Letter 0]

9 Mar 2020

PONE-D-20-03526

An iron-dependent metabolic vulnerability underlies VPS34-dependence in RKO cancer cells

PLOS ONE

Dear Dr Menon,

Thank you for submitting your manuscript to PLOS ONE. After careful consideration, we feel that it has merit but does not meet PLOS ONE’s publication criteria as it currently stands. Therefore, we invite you to submit a revised version of the manuscript that addresses the points raised during the review process.

Although the overall recommendation of the two reviewers differed substantially, both raise significant concerns. As such, if the authors resubmit it is likely that I will send out the manuscript for a third opinion at that stage. The authors only need to respond to comments that challenge their conclusions based on the experiments shown; I will not base my decision on how they could have used their large datasets in a more efficient manner, although I do of course invite them to consider these points and respond to the reviewer. I take it for granted that their data will be made available to all as a condition for publication of the manuscript.

We would appreciate receiving your revised manuscript by Apr 23 2020 11:59PM. To enhance the reproducibility of your results, we recommend that if applicable you deposit your laboratory protocols in protocols.io, where a protocol can be assigned its own identifier (DOI) such that it can be cited independently in the future. For instructions see: http://journals.plos.org/plosone/s/submission-guidelines#loc-laboratory-protocols

We look forward to receiving your revised manuscript.

Kind regards,

Fanis Missirlis, Ph.D.

Academic Editor

PLOS ONE

Journal Requirements:

2. We note that you are reporting an analysis of a microarray, next-generation sequencing, or deep sequencing data set. PLOS requires that authors comply with field-specific standards for preparation, recording, and deposition of data in repositories appropriate to their field. Please upload these data to a stable, public repository (such as ArrayExpress, Gene Expression Omnibus (GEO), DNA Data Bank of Japan (DDBJ), NCBI GenBank, NCBI Sequence Read Archive, or EMBL Nucleotide Sequence Database (ENA)). In your revised cover letter, please provide the relevant accession numbers that may be used to access these data. For a full list of recommended repositories, see http://journals.plos.org/plosone/s/data-availability#loc-omics or http://journals.plos.org/plosone/s/data-availability#loc-sequencing.

3. Thank you for providing the following Funding Statement: 

"Novartis funded the research contained in this manuscript. The funders had no role in study design, data collection and analysis, decision to publish, or preparation of the manuscript."

We note that one or more of the authors is affiliated with the funding organization, indicating the funder may have had some role in the design, data collection, analysis or preparation of your manuscript for publication; in other words, the funder played an indirect role through the participation of the co-authors.

If the funding organization did not play a role in the study design, data collection and analysis, decision to publish, or preparation of the manuscript and only provided financial support in the form of authors' salaries and/or research materials, please review your statements relating to the author contributions, and ensure you have specifically and accurately indicated the role(s) that these authors had in your study in the Author Contributions section of the online submission form. Please make any necessary amendments directly within this section of the online submission form.  Please also update your Funding Statement to include the following statement: “The funder provided support in the form of salaries for authors [insert relevant initials], but did not have any additional role in the study design, data collection and analysis, decision to publish, or preparation of the manuscript. The specific roles of these authors are articulated in the ‘author contributions’ section.”

If the funding organization did have an additional role, please state and explain that role within your Funding Statement.

Please also provide an updated Competing Interests Statement declaring this commercial affiliation along with any other relevant declarations relating to employment, consultancy, patents, products in development, or marketed products, etc.  

Reviewers' comments:

Reviewer's Responses to Questions

**Comments to the Author**

1. Is the manuscript technically sound, and do the data support the conclusions?

Reviewer #1: No

Reviewer #2: Yes

2. Has the statistical analysis been performed appropriately and rigorously? 

Reviewer #1: Yes

Reviewer #2: Yes

3. Have the authors made all data underlying the findings in their manuscript fully available?

Reviewer #1: Yes

Reviewer #2: No

4. Is the manuscript presented in an intelligible fashion and written in standard English?

Reviewer #1: Yes

Reviewer #2: Yes

5. Review Comments to the Author

Reviewer #1: In this manuscript, the authors identify a cancer cell line (RKO cells) as a VPS34-dependent cell line through RAB7A. They try to propose that VPS34 induces RAB7A-dependent lysosomal degradation of transferrin receptors, which led to impaired iron uptake. Upon several experiments, they suggest that iron limitation is a primary driver of VPS34-dependency in RKO cells. Unfortunately, this paper is plenty of correlations that are not supported at all by direct experimental evidence supporting the author’s claims. Also, what is the biological interest of demonstrating that in an in-vitro system, using only one cell line these findings are relevant for cancer? Below I have listed some major concerns:

MAJOR CONCERNS

• The authors use the shRNA of VPS34 and different members of C1 and CII to claim that CII is responsible for RKO sensitivity. However, they did not show direct evidence excluding CI from this phenotype. ATG14 depletion is not enough since all the common members of CI and CII are involved. Block autophagy from so early steps would have some effects, I guess.

• RAB7A is just one (the most significant, but there are a lot of significant hits) of the potential downstream effectors of VPS34. Why the authors focus on it. As expected, and showed by the authors, RAB7A KO partially mimics VPS34 inhibitions. What is the reason to use very sophisticated screening methods to focus on one protein that even does not fully recapitulate the phenotype? This is a mix of results linked together without a strong rationale….

• There are no evidences in the manuscript indicating that VPS34 induces RAB7A!!

• Then, the authors come back to omics, using a hit that is not strongly supported by experimental evidence, performing transcriptomics analysis. From this analysis, GO indicates that cholesterol pathway and HIF1 are significantly modulated in RAB7A. Cholesterol pathway is discarded without check cholesterol levels in cells? What is the rationale to add cholesterol to the cells, why not deplete cholesterol? Again, no clear rationale …

• Finally, the authors decide to focus on iron homeostasis. IRP2 levels and aconitase activity are an indirect approach to evaluate the intracellular iron availability... why they didn’t look at the quantification of the labile iron pool (LIP) and ferritin protein levels (whose expression, as TfR, is controlled by IRPs)?

• The staining of the Transferrin Alexa Fluor 488 probe in WT and RAB7A KO cells is strange. In particular, the group RAB7A KO treated with a VPS34 inhibitor. Does that mean the positive vacuoles positive (in their limiting membranes) for the probe in this group?

Comments:

• Moreover, to me, it is necessary to check at ferritin because the inhibition of VPS34 by PIKIII also caused impairment in ferritinophagy, as described in the first paper that they cited. They only observed that RCO cells are not suffering by depletion of ferritinophagy (fig. S3), but they didn’t show the ferritin levels in Ctrl and RAB7a KO cells during PIKIII treatment (i.e. in figure 6F-G)

• As VPS34 inhibition provokes the shift from oxidative phosphorylation towards glycolysis, the luciferase assay (CellTiter-Glo) is a reliable readout of cell viability?

• How do the authors explain that at a higher dose of PIKIII the increment of HIF1 appears also in RAB7 KO cells (Fig. 3D)?

Reviewer #2: The manuscript by Kobylarz and colleaguesexplores the dependency of RKO cancer cells on the protein Vps34. The experiments are well conducted and presented, and the story line flows logically.

The authors show, using a KO screen, that in the absence of Vps34 activity, removal of Rab7a is sufficient for viability. They then search for the underlying mechanism using a transcriptome approach to identify the underlying mechanism. There are, however, a few key experiments that are required in order to support the conclusions drawn by the authors.

- The transcriptional signature of Rab7A-KO+Vps34 inhibition pointed towards activation of HIF-1a and cholesterol synthesis. This is exactly the same signature that is obtained by inhibiting lysosomal acidification (see for example 31983508, 28296633, 31793879, which the authors anyway cite). Yet, the authors never test this possibility. In my view, this experiment is required, because it raises the possibility that the Vps34-Rab7A pathway acts by a completely different mechanism than that suggested by the authors. This is further substantiated by the fact that the authors can rescue many of the observed cellular and molecular phenotypes with exogenous iron citrate.

- line 229 - the statement that bioavailable iron is predominantly complexed with transferrin is not necessarily correct. In fact, upon iron deficiency, cells readily activate ferritinophagy and mitophagy, as well as up-regulation of the transferrin receptor. Therefore, it is pivotal that the autophagy branch of the pathway is also tested. I recall that the authors showed that the removal of Vps34-C1 had no effect on the viability, while C2 did, which they possibly used to discount the involvement of autophagy. However, this is not sufficient, as UVRAG, involved in C2, is also involved in autophagy. Thus, it is important that the authors test if the autophagy pathway is involved (either generally, by removing a protein needed for autophagosome formation such as Atg5 or Atg7, or by specifically ablating ferritinophagy via NCOA4).

- line 252 - the use of bafilomycin is per se sufficient to trigger iron deficiency and TFRC induction (see papers mentioned above in the context of HIF-1a). Therefore this experiment should be done in the presence of a lysosomal inhibitor that does not affect lysosomal pH - for example, leupeptin or some other caspase inhibitor.

- in Suppl Fig2, it is unclear if the slight differences on OCR are significant (they don't seem to be) - it should be noted whatever the case is. Also, Rab7A should be compared to the controls, i.e., in the same plate and not just in terms of absolute OCR.

- the analysis of the RNA seq data is unnecessarily conservative. Behjamin-Hochberg is known to hammer the data. That is the authors' choice, so I will not dwell on it. However, the RNA seq raw data files should be made available in a repository (e.g., GEO database), so that other interested labs can reanalyze the data with statistical strategies that are more appropriate for biological data.

6. PLOS authors have the option to publish the peer review history of their article (what does this mean?). If published, this will include your full peer review and any attached files.

Reviewer #1: No

Reviewer #2: No

---

## [Author Response · Author response to Decision Letter 0]

29 May 2020

Reviewer #1

In this manuscript, the authors identify a cancer cell line (RKO cells) as a VPS34-dependent cell line through RAB7A. They try to propose that VPS34 induces RAB7A-dependent lysosomal degradation of transferrin receptors, which led to impaired iron uptake. Upon several experiments, they suggest that iron limitation is a primary driver of VPS34-dependency in RKO cells. Unfortunately, this paper is plenty of correlations that are not supported at all by direct experimental evidence supporting the author’s claims. Also, what is the biological interest of demonstrating that in an in-vitro system, using only one cell line these findings are relevant for cancer? Below I have listed some major concerns:

• Inhibition of VPS34 inhibits growth of most cancer cell lines, but to various degrees as visualized in figure 1. By computationally screening for genetic or expression features, we were unable to identify general predictors of VPS34 sensitivity across all cell lines. This is the reason why we decided to focus on one highly VPS34-dependent cell line and perform an in depth analysis what drives dependency. This is acknowledged in the title of our manuscript and we do not generalize our findings.

• We disagree that our paper is plenty of correlations not supported by direct experimental evidence. Using rescue experiments, we demonstrate that iron is necessary and sufficient to reverse both RKO growth defect and metabolic alterations due to VPS34 inhibition.

• Furthermore, we expect the pooled CRISPR screen, transcriptional profiling and lipidomics data sets to be a valuable resource for future studies on VPS34 biology.

MAJOR CONCERNS

The authors use the shRNA of VPS34 and different members of C1 and CII to claim that CII is responsible for RKO sensitivity. However, they did not show direct evidence excluding CI from this phenotype. ATG14 depletion is not enough since all the common members of CI and CII are involved. Block autophagy from so early steps would have some effects, I guess.

• We have included project DRIVE data (S1A Fig) and individual dox-inducible shRNAs (S1B and S1C Figs) showing that knockdown of canonical autophagy genes ATG5, ATG7 and ATG16L1 do not significantly impact growth of RKO cells 

RAB7A is just one (the most significant, but there are a lot of significant hits) of the potential downstream effectors of VPS34. Why the authors focus on it. As expected, and showed by the authors, RAB7A KO partially mimics VPS34 inhibitions. What is the reason to use very sophisticated screening methods to focus on one protein that even does not fully recapitulate the phenotype? This is a mix of results linked together without a strong rationale….

• The goal of the pooled CRISPR screen was to map the cellular pathways that can suppress VPS34-dependency, without a comprehensive follow-up of all hits. We have clarified this in the manuscript. 

• The pooled CRISPR screen identified a strong enrichment of suppressors that regulate endolysosomal maturation. In addition to being the strongest hit, RAB7A was picked based on its well characterized role in endolysosomal maturation. We view the RAB7A KO data as validation of endolysosomal maturation being a key driver of VPS34-depenency in RKO cells. To better reflect this point, he have re-organized the manuscript and now show all RAB7A KO data in the final figure, linking it to transferrin receptor trafficking, iron uptake, metabolic alterations and cell growth.

• The partial rescue in RAB7 KO cells suggests that the VPS34-RAB7A axis plays a critical role, but is likely not the only contributor to the VPS34-dependent growth defect. Involvement of other pathways such as sphingolipid metabolism will be explored in further studies.

• The loss of RAB7A alone has a partial iron deprivation effect, which is reversed by PIK-III treatment suggesting a complex relationship between RAB7A and VPS34 in the endosome maturation process and regulation of iron homeostasis.

There are no evidences in the manuscript indicating that VPS34 induces RAB7A!!

• We acknowledge that the manuscript lacks experimental evidence that VPS34 inhibition induces RAB7A. Given the scope of this study, we focused on characterizing the iron-dependency rather than the molecular mechanism how VPS34 inhibition activates RAB7A. Unfortunately, we are unable to deliver such data now due to the COVID19-related lockdown and can only refer in the discussion to Jaber et al. who have shown that VPS34 deficient MEFs accumulate GTP bound RAB7A due to feedback inhibition of TBC1D2, the GAP for RAB7A. 

Then, the authors come back to omics, using a hit that is not strongly supported by experimental evidence, performing transcriptomics analysis. From this analysis, GO indicates that cholesterol pathway and HIF1 are significantly modulated in RAB7A. Cholesterol pathway is discarded without check cholesterol levels in cells? What is the rationale to add cholesterol to the cells, why not deplete cholesterol? Again, no clear rationale …

• We have now included a complete analysis of the cholesterol deprivation phenotype in Figure 4. Using lipidomics and labeled LDL probe, we show that VPS34 inhibition decreases intracellular cholesteryl esters and blocks cholesterol uptake through enhanced lysosomal degradation of LDL receptors. Since PIK-III treated RKO cells display cholesterol deprivation phenotype, we added excess cholesterol to see if this could rescue the growth defect, which it did not.

Finally, the authors decide to focus on iron homeostasis. IRP2 levels and aconitase activity are an indirect approach to evaluate the intracellular iron availability... why they didn’t look at the quantification of the labile iron pool (LIP) and ferritin protein levels (whose expression, as TfR, is controlled by IRPs)?

• We evaluated colorimetric iron assay kits, but were unable to validate a reliable method to measure intracellular iron levels in RKO cells. Therefore, we settled on examining intracellular iron levels by indirect and established approaches such as aconitase activity assay and immunoblots of IRP2 levels.

• We have now included immunoblots for ferritin (FTH1) (S2A and S4B Figs). Ferritin levels are reduced at low doses of PIK-III, in line with cellular iron depletion. 

The staining of the Transferrin Alexa Fluor 488 probe in WT and RAB7A KO cells is strange. In particular, the group RAB7A KO treated with a VPS34 inhibitor. Does that mean the positive vacuoles positive (in their limiting membranes) for the probe in this group?

• VPS34 inhibition is known to cause enlargement of late endosomes and lysosomes (2), which we also observe in RKO cells treated with PIK-III. The transferrin probe binds to transferrin receptor found on the luminal side of the endosome limiting membrane hence, the positively stained limiting membranes seen in PIK-III treated RAB7A KO cells.

• We have examined the PIK-III-induced vacuoles in RKO cells in more detail and included the data here for full transparency (Fig R1). We only included the data for the response to reviewers because we believe it does not add significant value to the manuscript. Phase-contrast microscopy confirmed that VPS34 inhibition induced enlarged vacuoles in RKO CTRL cells, and RAB7A KO caused even greater and larger vacuoles (Fig R1A). Using confocal microscopy, we identified that the vacuoles in RKO RAB7A KO cells treated with PIK-III contain transferrin receptor in the limiting membrane (Fig R1B). 

Comments:

Moreover, to me, it is necessary to check at ferritin because the inhibition of VPS34 by PIKIII also caused impairment in ferritinophagy, as described in the first paper that they cited. They only observed that RCO cells are not suffering by depletion of ferritinophagy (fig. S3), but they didn’t show the ferritin levels in Ctrl and RAB7a KO cells during PIKIII treatment (i.e. in figure 6F-G)

• We would like to thank the reviewer for this suggestion, which has also been raised by reviewer #2. We have added immunoblots for ferritin (FTH1) and the ferritinophagy component NCOA4 (S2A and S4B Figs). We see minimal effects on NCOA4 and decreased ferritin levels, at low doses of PIK-III when we observe clearance of TFR. It is only at higher concentrations of PIK-III that we see inhibition of ferritinophagy, and accumulation of NCOA4 and ferritin. We have clarified this point in the manuscript starting on line 295 in the results section and on line 444 on the discussion section. 

As VPS34 inhibition provokes the shift from oxidative phosphorylation towards glycolysis, the luciferase assay (CellTiter-Glo) is a reliable readout of cell viability?

• We have assessed cell viability using both CellTiter-Glo (Fig 6C) and colony formation (Fig 6B) assays, which show similar results.

How do the authors explain that at a higher dose of PIKIII the increment of HIF1 appears also in RAB7 KO cells (Fig. 3D)?

• At higher doses of PIK-III, knockout of RAB7A does not fully rescue iron deprivation phenotype including IRP2 levels, OCR/ECR ratio and HIF1� accumulation. This is consistent with the fact that loss of RAB7A partially rescues the growth defect due to PIK-III treatment shifting the IC50 curve ~9-fold. 

Reviewer #2

The manuscript by Kobylarz and colleagues explores the dependency of RKO cancer cells on the protein Vps34. The experiments are well conducted and presented, and the story line flows logically.

The authors show, using a KO screen, that in the absence of Vps34 activity, removal of Rab7a is sufficient for viability. They then search for the underlying mechanism using a transcriptome approach to identify the underlying mechanism. There are, however, a few key experiments that are required in order to support the conclusions drawn by the authors.

Comments:

The transcriptional signature of Rab7A-KO+Vps34 inhibition pointed towards activation of HIF-1a and cholesterol synthesis. This is exactly the same signature that is obtained by inhibiting lysosomal acidification (see for example 31983508, 28296633, 31793879, which the authors anyway cite). Yet, the authors never test this possibility. In my view, this experiment is required, because it raises the possibility that the Vps34-Rab7A pathway acts by a completely different mechanism than that suggested by the authors. This is further substantiated by the fact that the authors can rescue many of the observed cellular and molecular phenotypes with exogenous iron citrate.

• We have now addressed impaired endosomal acidification as a possible mechanism. Using the pH-sensitive pHrodo Red and the pH-insensitive Alexa Fluor 488 transferrin probes we show that the intensities of the transferrin probes reduce with 24 hours of PIK-III treatment as expected but, the ratio of the probes do not change compared to vehicle treatment. This indicates that PIK-III does not impact the acidification of the transferrin-positive endosomes. Inhibition of endolysosomal acidification with bafilomycin A1 was used as a positive control (S3 Fig).

line 229 - the statement that bioavailable iron is predominantly complexed with transferrin is not necessarily correct. In fact, upon iron deficiency, cells readily activate ferritinophagy and mitophagy, as well as up-regulation of the transferrin receptor. Therefore, it is pivotal that the autophagy branch of the pathway is also tested. I recall that the authors showed that the removal of Vps34-C1 had no effect on the viability, while C2 did, which they possibly used to discount the involvement of autophagy. However, this is not sufficient, as UVRAG, involved in C2, is also involved in autophagy. Thus, it is important that the authors test if the autophagy pathway is involved (either generally, by removing a protein needed for autophagosome formation such as Atg5 or Atg7, or by specifically ablating ferritinophagy via NCOA4).

• We would like to thank the reviewer for these suggestions, which have been also raised by reviewer #1. 

• We have included project DRIVE data (S1A Fig) and individual dox-inducible shRNAs (S1B and S1C Figs) showing that knockdown of canonical autophagy genes ATG5, ATG7 and ATG16L1 do not significantly impact growth of RKO cells 

We have added immunoblots for the ferritinophagy component NCOA4 (S2A and S4B Figs). At low doses of PIK-III when we observe clearance of TFR, we see minimal effects on NCOA4. It is only at higher concentrations of PIK-III that we see inhibition of ferritinophagy and accumulation of NCOA4. We have clarified this point in the manuscript starting on line 295 in the results section and on line 444 in the discussion section.

line 252 - the use of bafilomycin is per se sufficient to trigger iron deficiency and TFRC induction (see papers mentioned above in the context of HIF-1a). Therefore this experiment should be done in the presence of a lysosomal inhibitor that does not affect lysosomal pH - for example, leupeptin or some other caspase inhibitor.

• Chloroquine has a different mechanism of action to bafilomycin A1 in blocking lysosomal activity and we show that chloroquine treatment also prevents PIK-III mediated reduction of LDLR and TFR (Fig 5H). 

• The pooled CRISPR screen also revealed cathepsin L (CTSL) as a PIK-III suppressor. Cathepsin L is a lysosomal endopeptidase involved in protein degradation. While we have not rigorously investigated the CTSL mechanism of action in our model, we are including additional cell viability data in this document for full transparency. Loss of CTSL (RKO CTSL KO), or inhibition with the selective CTSL inhibitor (Z-Phe-Tyr(tBu)-diazomethylketone; CTSLi) partially rescued cell proliferation in cells treated with PIK-III (Figure R2). This data suggests that inhibition of a single lysosomal protease is sufficient to partially rescue RKO cell growth upon VPS34 inhibition. 

in Suppl Fig2, it is unclear if the slight differences on OCR are significant (they don't seem to be) - it should be noted whatever the case is. Also, Rab7A should be compared to the controls, i.e., in the same plate and not just in terms of absolute OCR.

• We have included OCR and ECAR data for RKO CTRL and RAB7A KO cells in Fig S5. The seahorse experiments with CTRL and RAB7A KO cells (or DMSO and FAC treatment) were performed in the same plate to allow for direct comparison. We merely presented the data separately for ease of visualization. 

the analysis of the RNA seq data is unnecessarily conservative. Behjamin-Hochberg is known to hammer the data. That is the authors' choice, so I will not dwell on it. However, the RNA seq raw data files should be made available in a repository (e.g., GEO database), so that other interested labs can reanalyze the data with statistical strategies that are more appropriate for biological data.

• RNA-seq data was uploaded to the Sequence Read Archive. The accession number is PRJNA633293. Data will be released publicly upon manuscript acceptance.

---

## [Decision Letter · Decision Letter 1]

18 Jun 2020

An iron-dependent metabolic vulnerability underlies VPS34-dependence in RKO cancer cells

PONE-D-20-03526R1

Dear Dr. Menon,

We’re pleased to inform you that your manuscript has been judged scientifically suitable for publication and will be formally accepted for publication once it meets all outstanding technical requirements.

I would like to congratulate you for a very nice study that contributes greatly to the connection between membrane trafficking, cholesterol and iron biology.

Kind regards,

Fanis Missirlis, Ph.D.

Academic Editor

PLOS ONE

Additional Editor Comments (optional):

Reviewers' comments:

Reviewer's Responses to Questions

**Comments to the Author**

1. If the authors have adequately addressed your comments raised in a previous round of review and you feel that this manuscript is now acceptable for publication, you may indicate that here to bypass the “Comments to the Author” section, enter your conflict of interest statement in the “Confidential to Editor” section, and submit your "Accept" recommendation.

Reviewer #1: All comments have been addressed

Reviewer #2: All comments have been addressed

2. Is the manuscript technically sound, and do the data support the conclusions?

Reviewer #1: Yes

Reviewer #2: Yes

3. Has the statistical analysis been performed appropriately and rigorously? 

Reviewer #1: Yes

Reviewer #2: Yes

4. Have the authors made all data underlying the findings in their manuscript fully available?

Reviewer #1: Yes

Reviewer #2: Yes

5. Is the manuscript presented in an intelligible fashion and written in standard English?

Reviewer #1: Yes

Reviewer #2: Yes

6. Review Comments to the Author

Reviewer #1: I appreciate the big effort the authors made to review this manuscript. They answered satisfactorily most of my major comments. Thanks

Reviewer #2: The authors have addressed my comments.

The authors have addressed my comments.

The authors have addressed my comments.

7. PLOS authors have the option to publish the peer review history of their article (what does this mean?). If published, this will include your full peer review and any attached files.

Reviewer #1: No

Reviewer #2: Yes: Nuno Raimundo